# A Progressive Evidence Localization Framework Based on Wasserstein Gradient Flows for Document Visual Question Answering

Haosen Wang [1]  Jing Xiao [2]  Mengqiao Li [1]  Xuanze Wang [1]  Mingzheng Zhang [1]  Xiaowang Zhang [1]  Zhiyong Feng [1]

## Abstract

Precise localization of evidence regions in Document Visual Question Answering is crucial for improving model interpretability and reliability. However, existing methods predominantly adopt one-step localization strategies, which often fail to effectively distinguish evidence regions from irrelevant content when page semantics are complex or evidence regions are extremely small, leading to ambiguous boundaries and inaccurate localization. To address this issue, we propose a progressive evidence localization framework based on Wasserstein gradient flow, which formulates evidence localization as an optimal transport problem over probability distributions. Since continuous gradient flows are intractable in practice, we employ the Jordan-Kinderlehrer-Otto (JKO) scheme for discrete optimization and further derive an end-to-end trainable loss function that transforms the theoretical formulation into a neural network optimization objective, enabling coarse-to-fine precise characterization of evidence regions. Experimental results demonstrate that the proposed method significantly outperforms existing approaches in both evidence localization and answer generation tasks.

## 1. Introduction

Document Visual Question Answering (DocVQA) (Qin et al., 2025; Mathew et al., 2021; Kim et al., 2022; Herbst et al., 2025) aims to generate answers by comprehending document images, while precise evidence localization requires the model to accurately identify the source regions

supporting its predictions. Such capability is crucial for improving model interpretability and reliability (Liu et al., 2025; Zhu et al., 2022; 2023), and further facilitates downstream applications such as intelligent retrieval (Cho et al., 2025; Wang et al., 2025), document editing (Suri et al., 2024), and information extraction (Hu et al., 2024).

To achieve precise evidence localization, early studies primarily relied on pretrained document understanding models to encode entire pages holistically, with representative approaches including the LayoutLM series (Xu et al., 2020; 2021; Huang et al., 2022). However, due to the lack of explicit modeling of evidence sources, their reasoning processes remain difficult to interpret. To alleviate this limitation, recent studies have introduced explicit evidence localization mechanisms. EaGERS (Hormazábal et al., 2025) proposes an interpretable pipeline that maps natural language reasoning to page subregions by computing multimodal embedding similarities over configurable grids and incorporating majority voting, thereby constraining answer generation. DocVXQA (Souibgui et al., 2025) further formulates interpretability as a learning objective, leveraging visual attention heatmaps to highlight critical regions while improving the traceability of predictions without sacrificing answer accuracy.

However, as illustrated in Figure 1, most existing approaches adopt a one-step localization paradigm that directly maps from the entire page to evidence regions, lacking a progressive coarse-to-fine focusing process. When document pages contain complex semantic structures or evidence regions occupy only a very small portion of the page, these methods often struggle to effectively distinguish relevant evidence from irrelevant content, resulting in ambiguous boundaries, inaccurate localization, or even missing critical evidence, which consequently undermines the reliability and interpretability of generated answers.

To address the above challenges, we propose a progressive evidence localization framework based on Wasserstein gradient flow (Ambrosio et al., 2005; Friesecke, 2024), which reformulates precise evidence localization as an optimal transport problem over probability distributions (Proposition 3.5). Specifically, the distribution of candidate evidence

[1]School of Computer Software, Tianjin University, Tianjin, China [2]International Engineering Institute, Tianjin University, Tianjin, China. Correspondence to: Haosen Wang <haosenwang@tju.edu.cn>.

*Proceedings of the 43rd International Conference on Machine Learning*, Seoul, South Korea. PMLR 306, 2026. Copyright 2026 by the author(s).

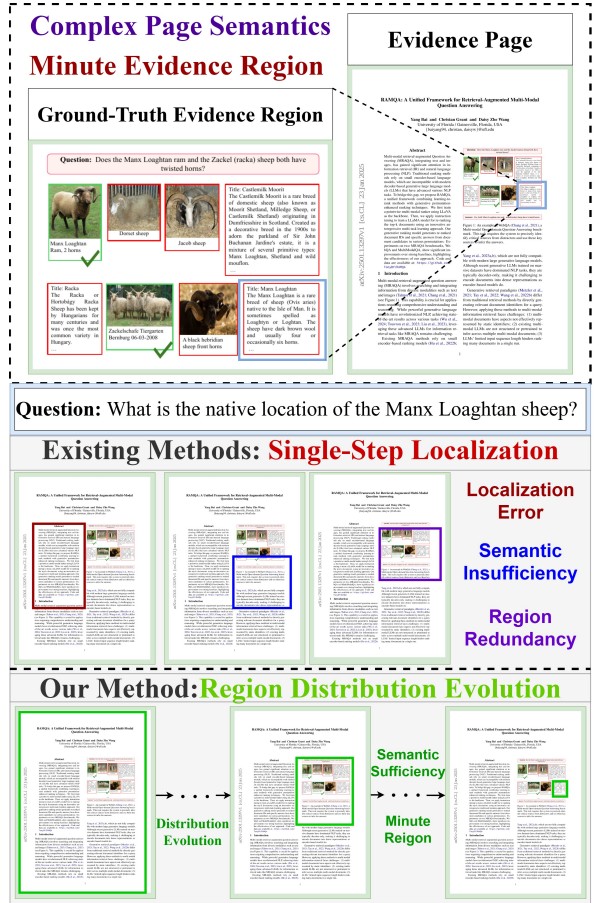

*Figure 1.* Existing single-step localization methods struggle to handle scenarios with complex page semantics or extremely small evidence regions; in contrast, our approach achieves precise localization and interpretable reasoning through progressive evolution.

regions is progressively evolved from an initial distribution, where evidence is assumed to be spread across the entire page, toward a target distribution concentrated on the true evidence regions, thereby enabling coarse-to-fine precise characterization of evidence regions.

Since continuous-time gradient flows are intractable to solve directly in practice, we adopt the Jordan–Kinderlehrer–Otto (JKO) (Jordan et al., 1998) scheme to perform discrete optimization of the gradient flow, decomposing the continuous evolution process into iterative distribution updates over multiple discrete time steps. More importantly, based on our theoretical derivation, we obtain an end-to-end trainable loss function (Corollary 3.12), which transforms the theoretical framework into an objective that can be directly optimized by neural networks.

Furthermore, at each evolution step, we propose a semantic density updater to parameterize the semantic density of candidate regions by decomposing it into intrinsic semantic density and conditional semantic density. The former provides a filtering prior based on the inherent semantic characteristics of each region, while the latter dynamically focuses on evidence regions according to the question semantics. Their collaboration enables the evolution process to simultaneously achieve semantic stability and question-guided adaptability.

The proposed framework progressively concentrates on critical evidence through progressive evolution, enabling precise evidence localization and interpretable reasoning. The main contributions of this paper are summarized as follows:

1. We propose a progressive evidence region localization framework based on Wasserstein gradient flows, enabling a precise coarse-to-fine characterization of evidence regions

2. We perform discrete optimization of the continuous-time gradient flow based on the JKO scheme and derive an end-to-end trainable loss function, thereby transforming the theoretical framework into an objective that can be directly optimized by neural networks.

3. Experimental results demonstrate that the proposed method significantly outperforms existing approaches in both evidence localization and answer prediction, while providing an interpretable progressive reasoning process.

## 2. Related Work

### 2.1. Precise Evidence Region Localization in Document Visual Question Answering

Precise evidence localization is a fundamental component for building trustworthy DocVQA systems. Early approaches primarily relied on implicit global encoding. Representative works such as the LayoutLM series (Xu et al., 2020; 2021; Huang et al., 2022) achieved remarkable performance on DocVQA through joint pretraining over text, layout, and visual modalities, while TILT (Powalski et al., 2021) further introduced layout-aware attention biases for answer decoding. However, these methods only treat attention weights as indirect evidence and therefore cannot explicitly trace the source regions supporting the generated answers. Recent studies have shifted toward explicit region-level supervision and interpretability objectives. Sci-EGQA (Yu et al., 2026) construct a large-scale benchmark with bounding-box annotations. EaGERS (Hormazábal et al., 2025) maps natural language reasoning to page sub-regions through multimodal embedding similarity, while DocVXQA (Souibgui et al., 2025) formulates contextual sufficiency and representation efficiency as explicit learning objectives, enabling self-explainable answer prediction via visual heatmaps. Nevertheless, these methods all adopt a one-step localization paradigm, which is prone to ambiguous boundaries and localization drift when evidence

regions occupy only a small portion of the page or when document semantics are densely distributed, due to the lack of a progressive coarse-to-fine focusing mechanism.

## 2.2. Wasserstein Gradient Flows

Wasserstein Gradient Flows (WGF) provide an optimization framework for probability distribution evolution. Their foundational idea can be traced back to the JKO variational scheme (Jordan et al., 1998), which first characterized the Fokker–Planck equation as the steepest descent flow of free energy in the space of probability measures. Subsequent work by Ambrosio *et al.* (Ambrosio et al., 2005) established a comprehensive theoretical foundation regarding existence and convergence, thereby significantly advancing the mathematical maturity of the framework. To overcome the computational challenges of continuous WGF in high-dimensional settings, Mokrov *et al.* (Mokrov et al., 2021) first introduced differentiable parameterization of JKO steps using input convex neural networks. Lambert *et al.* (Lambert et al., 2022) further proposed a variational inference algorithm with convergence guarantees based on Bures–Wasserstein geometry, while Choi *et al.* (Choi et al., 2024) exploited the equivalence between JKO and unbalanced optimal transport to reduce the training complexity from $\mathcal{O}(K^2)$ to $\mathcal{O}(K)$. More recent studies have extended WGF to more sophisticated vision and generative tasks. For example, Wasserstein Flow Matching (Haviv et al., 2025) elevates flow matching to the space of distribution families for generating both 2D and 3D shapes. Collectively, these advances demonstrate that WGF has achieved sufficient theoretical rigor and practical scalability to support large-scale learning tasks.

## 3. Method

This section systematically presents the proposed progressive evidence localization framework based on Wasserstein gradient flow. We first provide a formal definition of the problem in Section 3.1. We then design an energy functional satisfying both semantic sufficiency (Definition 3.1) and spatial minimality (Definition 3.2) constraints in Section 3.2, thereby reformulating precise evidence localization as a Wasserstein gradient flow optimization problem. Since continuous-time gradient flows are difficult to solve directly in practice, we further adopt the JKO scheme to discretize the gradient flow and derive a loss function that can be directly optimized by neural networks in Section 3.3. In Section 3.4, we introduce a semantic density updater that progressively updates the semantic density of candidate regions at each evolution step. Finally, as described in Section 3.5, the entire framework progressively concentrates on critical evidence through iterative evolution, enabling precise evidence localization and interpretable reasoning.

### 3.1. Problem Formulation

Given a document $D$, an evidence page $P^* \in D$, and a question $Q$, precise evidence localization aims to learn a mapping function $f_\theta$ that jointly predicts the answer $A$ and the set of evidence regions $B^* \subset P^*$: $f_\theta(P^*, Q) = (A, B^*)$. Here, $B^* = \{B_i^*\} = \{(x_i^{\min}, y_i^{\min}, x_i^{\max}, y_i^{\max})\}$ denotes the bounding box of an evidence region. The predicted evidence region $R$ is expected to satisfy the following two properties as much as possible:

**Definition 3.1** (Semantic Sufficiency). A region $R$ is semantically sufficient with respect to the question $Q$ and answer $A$ if and only if $R$ preserves all the information in $P^*$ relevant to the answer $A$:

$$I(R; A \mid Q) = I(P^*; A \mid Q) \tag{1}$$

where $I(\cdot; \cdot)$ denotes mutual information.

**Definition 3.2** (Spatial Minimality). Among all regions satisfying semantic sufficiency, $R$ is spatially minimal if and only if:

$$R = \arg \min_{R' \subseteq P^*} \mathcal{A}(R') \text{ s.t. } I(R'; A \mid Q) = I(P^*; A \mid Q) \tag{2}$$

where $\mathcal{A}(\cdot)$ denotes the area.

Accordingly, precise evidence localization can be formulated as the following optimization problem:

$$\min \ \mathcal{A}(R) \quad \text{s.t.} \quad I(R; A \mid Q) = I(P^*; A \mid Q) \tag{3}$$

the objective simultaneously enforces semantic sufficiency of the evidence region, ensuring that no answer-relevant information is lost, and spatial minimality, ensuring that redundant regions are excluded as much as possible.

### 3.2. Energy Functional Design

For the optimization problem in Eq. (3), the conditional mutual information $I(R; A \mid Q)$ involves high-dimensional distribution integrals that are intractable to solve analytically, making conventional optimization methods difficult to apply. Therefore, we reformulate the problem as a Wasserstein gradient flow optimization problem in the space of probability measures. Specifically, the candidate region distribution evolves through gradient flow from an initial distribution $\mu^{(0)}$, where evidence is assumed to be distributed over the entire page, toward a target distribution $\mu^*$, where evidence is concentrated on the true evidence regions (Definition 3.3).

**Definition 3.3** (Candidate Region Distribution). The candidate region distribution $\mu \in \mathcal{P}(\Omega^*)$ is a probability measure defined over the evidence page space $\Omega^*$, where $\mathcal{P}(\Omega^*)$ denotes the space of all probability measures on $\Omega^*$. If $\mu$ is

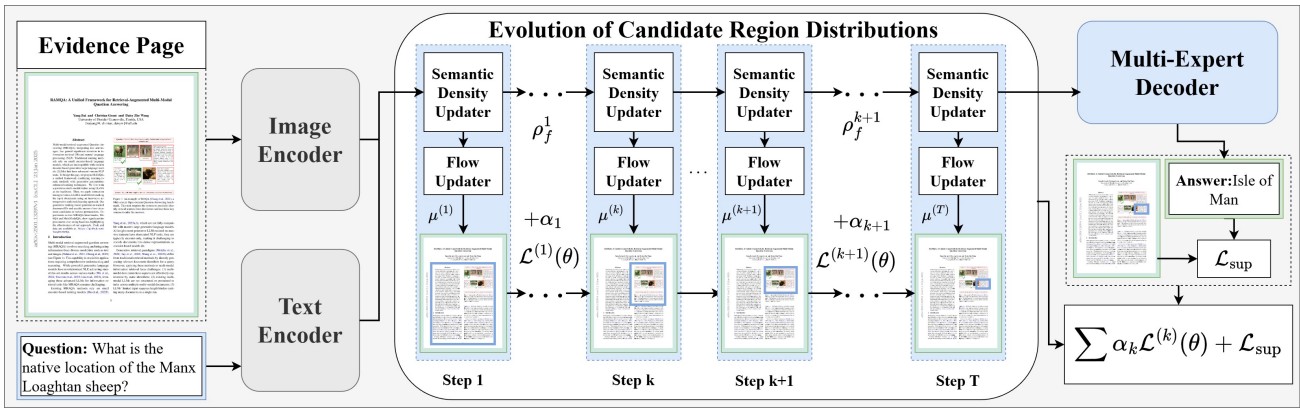

*Figure 2.* The progressive evidence region localization framework based on wasserstein gradient flows

absolutely continuous with respect to the Lebesgue measure $\mathcal{L}$, then there exists a density function $\rho$ such that

$$d\mu(x) = \rho(x)\,dx \qquad (4)$$

where $\rho(x)$ denotes the probability density that location $x$ contains evidence.

To apply Wasserstein gradient flow (Definition A.2) to evidence region localization, we design an energy functional to encode semantic sufficiency and spatial minimality.

**Definition 3.4** (Evidence Potential Function). The evidence potential function $V : \Omega^* \times \mathcal{Q} \times \mathcal{A} \to \mathbb{R}$ quantifies the evidence relevance of location $x$ with respect to the question $Q$ and answer $A$:

$$V(x; Q, A) = -\log p(A \mid x, Q) \qquad (5)$$

where $p(A \mid x, Q)$ denotes the conditional probability, and a smaller value of $V$ indicates stronger relevance.

The energy functional is formulated as:

$$\mathcal{E}[\mu; Q, A] = \underbrace{\int_{\Omega^*} V(x; Q, A)\,d\mu(x)}_{\mathcal{E}_{\text{pot}}[\mu]} - \underbrace{\lambda \int_{\Omega^*} \rho(x)\log\rho(x)\,dx}_{\mathcal{E}_{\text{ent}}[\mu]}$$
$$+ \underbrace{\gamma \mathcal{W}_2^2(\mu, \mu^*)}_{\mathcal{E}_{\text{trans}}[\mu]} \qquad (6)$$

where $\mathcal{E}_{\text{pot}}[\mu]$ denotes the potential energy term that drives the distribution toward regions with high evidence relevance; $\mathcal{E}_{\text{ent}}[\mu]$ denotes the entropy regularization term that encourages the distribution to become more concentrated, thereby achieving spatial minimality; $\mathcal{E}_{\text{trans}}[\mu]$ denotes the transport control term that guides the distribution toward the ground-truth evidence distribution $\mu^*$; and $\lambda, \gamma > 0$ are regularization parameters.

## 3.3. End-to-End Trainable Loss Function

Based on the energy functional (6), we reformulate the evidence region localization problem as a Wasserstein gradient flow optimization problem:

**Proposition 3.5.** *Let $\Omega^* \subseteq \mathbb{R}^d$ be a compact convex set, and let $\mu^{(0)}, \mu^* \in \mathcal{P}(\Omega^*)$ denote the initial distribution and the target distribution, respectively, both of which are absolutely continuous with respect to the Lebesgue measure. Assume that the potential function $V(x; Q, A)$ is $\lambda_V$-convex with respect to $x$ on $\Omega^*$ ($\lambda_V \geq 0$), and that $\lambda, \gamma > 0$.*

*Then, the evidence localization problem in Eq. (3) can be reformulated as solving the following Wasserstein gradient flow:*

$$\frac{\partial \mu_t}{\partial t} = \nabla \cdot \left( \mu_t \nabla \frac{\delta \mathcal{E}}{\delta \mu}[\mu_t] \right) \qquad (7)$$

$$= \nabla \cdot (\mu_t \nabla [V(x; Q, A)$$
$$- \lambda(1 + \log \rho_t(x)) + 2\gamma \phi(x)]) \qquad (8)$$

*where $\mu_t = \rho_t\,dx$ with $\rho_{t=0} = \rho^{(0)}$, and $\phi$ denotes the Kantorovich potential from $\mu_t$ to $\mu^*$. As $t \to \infty$, $\mu_t$ converges to the unique minimizer $\bar{\mu}$ of $\mathcal{E}$, whose support set $R^*$ corresponds to the evidence region satisfying both semantic sufficiency and spatial minimality.*

Although Proposition 3.5 establishes the theoretical foundation, the continuous gradient flow in Eq. (8) is difficult to solve directly. We adopt the Jordan–Kinderlehrer–Otto (JKO) scheme to discretize it and transform the continuous evolution into a stepwise optimization problem:

**Definition 3.6** (JKO Scheme). Given a time step $\tau > 0$, the JKO scheme is defined as:

$$\mu^{(k+1)} = \arg \min_{\mu \in \mathcal{P}(\Omega^*)} \left\{ \frac{1}{2\tau} \mathcal{W}_2^2(\mu, \mu^{(k)}) + \mathcal{E}[\mu; Q, A] \right\} \qquad (9)$$

where $\mu^{(k)}$ denotes the distribution at the $k$-th step.

Lemma A.4 and Proposition A.5 guarantee that, as $\tau \to 0$, the discrete sequence $\{\mu^{(k)}\}$ converges to the continuous gradient flow $\{\mu_t\}$.

**Definition 3.7** (Parameterized Density Function). Let $f_\theta : \Omega^* \times \mathcal{Q} \to \mathbb{R}$ be a neural network parameterized by $\theta \in \Theta$. The parameterized density is defined as:

$$\rho_\theta(x) = \frac{\exp(-f_\theta(x, Q))}{Z_\theta(Q)} \tag{10}$$

where $Z_\theta(Q) = \int_{\Omega^*} \exp(-f_\theta(x, Q)) \, dx$ and the corresponding candidate region distribution is denoted by $\mu_\theta(x)$. During training, the neural network $f_\theta$ learns to approximate the potential function $f_\theta(x, Q) \approx V(x; Q, A) = -\log p(A \mid x, Q)$

**Proposition 3.8** (Computable Optimization Objective). *Under neural network parameterization, the k-th JKO update in Eq. (9) is equivalent to minimizing the following loss function:*

$$\mathcal{L}^{(k)}(\theta) = \underbrace{\mathbb{E}_{x \sim \mu_\theta}[f_\theta(x, Q)]}_{\mathcal{L}_{pot}(\theta)} + \underbrace{\frac{1}{2\tau} \hat{\mathcal{W}}_2^2(\mu_\theta, \mu^{(k)})}_{\mathcal{L}_{trans}^{(k)}(\theta)}$$
$$+ \lambda \underbrace{\left(\mathbb{E}_{x \sim \mu_\theta}[f_\theta(x, Q)] + \log Z_\theta(Q)\right)}_{\mathcal{L}_{ent}(\theta)}$$
$$+ \gamma \underbrace{\hat{\mathcal{W}}_2^2(\mu_\theta, \mu^*)}_{\mathcal{L}_{prior}(\theta)} \tag{11}$$

*where $\mathcal{L}_{pot}$ denotes the potential energy term that enforces semantic sufficiency by encouraging localization toward highly evidence-relevant regions; $\mathcal{L}_{ent}$ denotes the entropy term that promotes distribution concentration, thereby achieving spatial minimality; $\mathcal{L}_{prior}$ constrains the distribution to evolve toward the target distribution; and $\mathcal{L}_{trans}^{(k)}$ denotes the JKO transport penalty term. Here, $\hat{\mathcal{W}}_2^2$ represents a differentiable approximation of the Wasserstein distance (Cuturi, 2013).*

Neural network parameterization of the potential function inevitably introduces approximation error, which propagates to the minimizer of the energy functional and consequently affects convergence accuracy. To characterize this effect, we present the following proposition.

**Proposition 3.9** (Energy Error Induced by Potential Approximation). *Let the ground-truth potential function be $V^*(x; Q, A) = -\log p(A \mid x, Q)$, and let its neural approximation be $f_\theta(x, Q)$. Define the approximation error as $\epsilon_{approx} = \sup_{x \in \Omega^*} |V_\theta(x) - V^*(x)|$ Let $\mathcal{E}$ and $\tilde{\mathcal{E}}$ denote the energy functionals constructed using $V^*$ and $V_\theta$, respectively, while sharing the same remaining terms. Let $\bar{\mu}$ and $\tilde{\mu}$ denote the unique minimizers of $\mathcal{E}$ and $\tilde{\mathcal{E}}$ over $\mathcal{P}(\Omega^*)$. Furthermore, let $\{\mu^{(k)}\}$ denote the sequence generated by the JKO iteration using $\tilde{\mathcal{E}}$. Then, the gap between the true*

energy and the optimal energy is linearly bounded by the approximation error:

$$\lim_{T \to \infty} \mathcal{E}[\mu^{(T)}] - \mathcal{E}[\bar{\mu}] \leq 2 \, \epsilon_{approx} \tag{12}$$

**Proposition 3.10** (Cumulative Error of Parameterized JKO). *Let $\{\mu_\theta^{(k)}\}_{k=0}^T$ denote the parameterized JKO sequence and $\{\mu_*^{(k)}\}_{k=0}^T$ denote the exact JKO sequence, both initialized from the same distribution $\mu^{(0)}$. Assume that the parameterization error at each step is uniformly bounded by $\delta_{param} = \sup_k \delta_{param}^{(k)}$, Then,*

$$\mathcal{W}_2(\mu_\theta^{(T)}, \mu_*^{(T)}) \leq C_1 \cdot \frac{1 - q^T}{1 - q} \delta_{param} \tag{13}$$

*where $q = \frac{1}{\sqrt{1 + \kappa\tau}} < 1$ is the contraction factor. Furthermore, the steady-state error is bounded as:*

$$\limsup_{T \to \infty} \mathcal{W}_2(\mu_\theta^{(T)}, \mu_*^{(T)}) \leq \frac{C_1}{1 - q} \delta_{param} \tag{14}$$

**Corollary 3.11** (End-to-End Error Bound). *Let $\bar{\mu}$ be the minimizer of the true energy functional $\mathcal{E}$ defined using $V^*$, and let $\{\mu_\theta^{(k)}\}$ denote the parameterized JKO sequence obtained by replacing $V^*$ with $V_\theta = f_\theta$. Under the assumptions of Propositions 3.9 and 3.10, we have*

$$\limsup_{T \to \infty} \mathcal{W}_2(\mu_\theta^{(T)}, \bar{\mu}) \leq \sqrt{\frac{4\epsilon_{approx}}{\kappa}} + \frac{C_1}{1 - q} \delta_{param} \tag{15}$$

Propositions 3.9 and 3.10, together with Corollary 3.11, demonstrate that parameterization errors propagate linearly to the final localization accuracy. Consequently, improving the expressive capacity of the neural network can further enhance localization quality.

To enable end-to-end optimization for both evidence localization and answer generation, we design a joint supervision loss $\mathcal{L}_{sup}$.

The evidence localization loss employs the KL divergence to constrain the predicted distribution $\mu^{(T)}$ to approximate the ground-truth distribution $\mu^*$:

$$\mathcal{L}_{reg} = \mathrm{KL}(\mu^* \| \mu^{(T)}) \tag{16}$$

The answer generation loss adopts an autoregressive objective conditioned on the localized evidence region $R_{\mu^{(T)}}$ and the question $Q$:

$$\mathcal{L}_{gen} = -\sum_{t=1}^{|y|} \log p(y_t \mid y_{<t}, R_{\mu^{(T)}}, Q) \tag{17}$$

The joint supervision loss is defined as:

$$\mathcal{L}_{sup} = \beta \cdot \mathcal{L}_{reg} + \eta \cdot \mathcal{L}_{gen} \tag{18}$$

where $\beta, \eta > 0$ are balancing coefficients.

**Corollary 3.12** (End-to-End Training Loss). *Let $T$ denote the total number of flow steps. The complete end-to-end training objective is defined as:*

$$\mathcal{L}_{total}(\theta) = \sum_{k=0}^{T-1} \alpha_k \mathcal{L}^{(k)}(\theta) + \mathcal{L}_{sup} \qquad (19)$$

*where $\mathcal{L}^{(k)}(\theta)$ denotes the gradient flow evolution loss under the JKO scheme (Eq. 11), $\{\alpha_k\}_{k=0}^{T-1}$ are the weights assigned to each step satisfying $\sum_k \alpha_k = 1$, and $\mathcal{L}_{sup}$ denotes the supervision loss (Eq. 18).*

### 3.4. Semantic Density Updater

To enable end-to-end training and evolve the initial distribution $\mu^{(0)}$ toward the target distribution $\mu^*$, each evolution step consists of a **semantic density updater** $F_E^k$ and a **flow updater** $\mathcal{U}_\theta^k$. The former integrates intrinsic semantic density and conditional semantic density based on the information-theoretic chain rule (Definition A.8), while the latter performs a one-step iterative update of the candidate region distribution.

At the $k$-th iteration ($k \geq 1$), the **semantic density updater** $F_E^k$ takes the semantic density from the previous step, denoted as $\rho_f^k$, and extracts as well as fuses semantic features through the following three modules.

*(1) Intrinsic Semantic Density Extractor $E_P$.* This module captures region-inherent semantic features of candidate regions via self-attention:

$$\rho_p^{k+1} = E_P(\rho_f^k) = \mathcal{F}_N\big(\mathcal{M}(\rho_f^k, \rho_f^k, \rho_f^k)\big) \qquad (20)$$

*(2) Conditional Semantic Density Extractor $E_Q$.* This module models question-dependent relevance using cross-attention between the question and candidate regions:

$$\rho_q^{k+1} = E_Q(\rho_f^k, Q) = \mathcal{F}_N\big(\mathcal{M}(Q, \rho_f^k, \rho_f^k)\big) \qquad (21)$$

*(3) Semantic Fusion Module $E_F$.* This module fuses the intrinsic and conditional semantic densities to produce the updated semantic density:

$$\rho_f^{k+1} = E_F(Q, \rho_p^{k+1}, \rho_q^{k+1}) = \mathcal{F}_N\big(\mathcal{M}(Q, \rho_p^{k+1}, \rho_q^{k+1})\big) \qquad (22)$$

Here, $\mathcal{M}(\cdot)$ denotes the multi-head attention mechanism (Definition A.9), $\mathcal{F}_N(\cdot)$ is a feed-forward network (Definition A.11), and $Q$ represents the question embedding.

**Flow Updater** $\mathcal{U}_\theta^k$. Given $\rho_f^{k+1}$, the flow updater performs a one-step iterative update of the candidate region distribution:

$$\mu^{(k+1)} = \mathcal{U}_\theta^k(\rho_f^{k+1}, \mu^{(k)}) = \text{Softmax}\left(-\rho_f^{k+1} - \frac{1}{\tau}W^{(k)}\right) \qquad (23)$$

---

**Algorithm 1** Evidence Region Localization Framework: Training Phase

---

**Require:** Training set $\{(P_i^*, Q_i, B_i^*, A_i^*)\}_{i=1}^N$, hyperparameters $\lambda, \gamma, \tau, T, \beta, \eta$
**Ensure:** Optimized model parameters $\theta^*$
 1: Initialize model parameters $\theta$
 2: **for** epoch $= 1$ to $E_{\max}$ **do**
 3:     $\mathcal{L}_{batch} \leftarrow 0$
 4:     **for** $(P^*, Q, B^*, A^*) \in \mathcal{B}$ **do**
 5:         // **Feature extraction**
 6:         $\mathbf{p} \leftarrow \mathcal{E}_{vis}(P^*), \quad \mathbf{q} \leftarrow \mathcal{E}_{txt}(Q)$
 7:         // **Flow-based iterative refinement**
 8:         $\boldsymbol{\rho}_f^0 \leftarrow \mathbf{p}$
 9:         **for** $k = 0$ to $T - 1$ **do**
 10:           // **Semantic density update**
 11:           $\boldsymbol{\rho}_p^{k+1} \leftarrow E_P^k(\boldsymbol{\rho}_f^k)$
 12:           $\boldsymbol{\rho}_q^{k+1} \leftarrow E_Q^k(\mathbf{q}, \boldsymbol{\rho}_f^k)$
 13:           $\boldsymbol{\rho}_f^{k+1} \leftarrow E_F^k(\mathbf{q}, \boldsymbol{\rho}_p^{k+1}, \boldsymbol{\rho}_q^{k+1})$
 14:           $\boldsymbol{\mu}^{(k+1)} \leftarrow \mathcal{U}_\theta^k(\boldsymbol{\rho}_f^{k+1}, \boldsymbol{\mu}^{(k)})$
 15:           $\mathcal{L}_{batch} \leftarrow \mathcal{L}_{batch} + \alpha_k \mathcal{L}^{(k)}(\theta)$
 16:         **end for**
 17:         $A \leftarrow \mathcal{D}_\theta(\mathbf{q}, \boldsymbol{\rho}_f^T)$
 18:         // **Supervised losses**
 19:         $\mathcal{L}_{batch} \leftarrow \mathcal{L}_{batch} + \mathcal{L}_{sup}$
 20:         // **Gradient update**
 21:         $\theta \leftarrow \theta - \ell_r \nabla_\theta \mathcal{L}_{batch}$
 22:     **end for**
 23: **end for**
 24: **return** $\theta^* \leftarrow \theta$

---

where $W^{(k)}$ denotes the Wasserstein-2 distance field induced by the previous distribution $\mu^{(k)}$, and $\tau > 0$ is the time step size.

### 3.5. Overall Framework

We present the complete pipeline of the proposed precise evidence localization framework in Figure 2. Given an evidence page $P^*$ and a question $Q$, a pretrained visual encoder $\mathcal{E}_{vis}$ and a text encoder $\mathcal{E}_{txt}$ are first employed to extract the visual representation $\mathbf{p} = \mathcal{E}_{vis}(P^*)$ and the semantic representation $\mathbf{q} = \mathcal{E}_{txt}(Q)$, respectively.

Subsequently, the candidate region distribution evolves from the initial distribution $\mu^{(0)}$ to the target distribution $\mu^{(T)} = \mu^*$ through $T$ steps of Wasserstein gradient flow evolution. The initial distribution $\mu^{(0)}$ corresponds to the evidence distribution over the entire evidence page, while the target distribution $\mu^*$ corresponds to the distribution over the ground-truth evidence region $R^*$. During the $k$-th evolution step ($k = 0, \ldots, T-1$), the semantic density $\rho_f^k$ is iteratively updated through the semantic density updater $F_E^k$ and the flow updater $\mathcal{U}_\theta^k$, ultimately yielding $\rho_f^T$ and

*Table 1.* Overall performance comparison on DocVQA, NiM-Benchmark, and DUDE

| Method | DocVQA | | | | NiM-Benchmark | | | | DUDE | | | |
|---|---|---|---|---|---|---|---|---|---|---|---|---|
| | EM | F1 | ANLS | mAP@IoU | EM | F1 | ANLS | mAP@IoU | EM | F1 | ANLS | mAP@IoU |
| GAP | 28.5 | 38.2 | 45.3 | 56.7 | 24.3 | 35.1 | 41.2 | 32.5 | 26.8 | 38.4 | 46.9 | 42.8 |
| DLaVA | 38.6 | 50.2 | 58.4 | 69.5 | 36.2 | 49.6 | 57.1 | 44.8 | 37.1 | 51.2 | 59.7 | 56.9 |
| EaGERS | 32.4 | 44.1 | 52.3 | 62.3 | 28.6 | 40.8 | 47.6 | 37.9 | 30.7 | 43.5 | 52.4 | 48.6 |
| ARIAL | 35.8 | 47.3 | 55.6 | 66.8 | 32.5 | 45.3 | 52.8 | 41.4 | 34.2 | 48.1 | 56.3 | 53.2 |
| DocVXQA | 40.5 | 52.6 | 59.8 | 72.1 | 39.4 | 53.7 | 61.5 | 46.5 | 39.3 | 53.8 | 62.4 | 57.4 |
| **Ours** | **43.2** | **55.8** | **62.7** | **79.6** | **45.8** | **59.4** | **67.3** | **53.4** | **42.1** | **57.6** | **66.5** | **62.8** |

$\mu^{(T)}$. This process progressively refines evidence localization from coarse-grained regions to fine-grained evidence areas.

To generate high-quality answers, we design a multi-expert collaborative decoder $\mathcal{D}_\theta$, which jointly produces the answer $A$ conditioned on the final semantic density $\rho_f^T$ and the question representation $\mathbf{q}$.

The decoder consists of $N$ experts $\mathcal{E} = \{E_1, \ldots, E_N\}$, where each expert independently extracts interaction features between the question and the evidence regions:

$$\mathbf{h}_i = E_i(\mathbf{q}, \rho_f^T) = \mathcal{F}_N\big(\mathcal{M}(\mathbf{q}, \rho_f^T, \rho_f^T)\big) \qquad (24)$$

To improve efficiency, we introduce a Top-$K$ gating mechanism. A gating network $\mathcal{G}_\theta$ computes the relevance weight of each expert:

$$\mathbf{w} = \mathcal{G}_\theta(\mathbf{q}, \rho_f^T) = \mathcal{F}\big(\mathcal{M}(\mathbf{q}, \rho_f^T, \rho_f^T)\big) \qquad (25)$$

The Top-$K$ experts $\{i_1, \ldots, i_K\}$ with the highest weights are selected, and their weights are renormalized as

$$\tilde{w}_{i_j} = \frac{w_{i_j}}{\sum_{j'=1}^K w_{i_{j'}}} \qquad (26)$$

The final answer is generated by a weighted fusion of the selected expert representations:

$$A = \text{LM}_{\text{head}}\left(\sum_{j=1}^K \tilde{w}_{i_j} \cdot \mathbf{h}_{i_j}\right) \qquad (27)$$

This mechanism enables the model to dynamically select and integrate the most relevant expert knowledge, thereby producing accurate answers.

During training, we freeze the parameters of the pretrained visual encoder $\mathcal{E}_{\text{vis}}$ and text encoder $\mathcal{E}_{\text{txt}}$, and only optimize the semantic density updaters $\{F_E^k\}_{k=0}^{T-1}$ involved in the evolution process, together with the multi-expert collaborative decoder $\mathcal{D}_\theta$. In addition, we treat the step-wise evolution loss weights $\{\alpha_k\}_{k=0}^{T-1}$ as learnable parameters, allowing them to be adaptively adjusted during training to balance

the optimization objectives across different evolution steps. The complete training procedure is summarized in Algorithm 1.

## 4. Experiments

### 4.1. Experimental Setup

**Datasets.** We evaluate the proposed method on three benchmark datasets: (1) NiM-Benchmark (Thakkar et al., 2025), which consists of multi-page documents where evidence regions exhibit spatially minimal distributions; (2) DUDE (Landeghem et al., 2023), which covers a wide variety of document types; and (3) DocVQA (Mathew et al., 2021), a large-scale benchmark for document visual question answering. Detailed dataset descriptions are provided in Appendix B.

**Baseline Methods.** We compare our method with five representative approaches. DocVXQA (Souibgui et al., 2025) performs localization based on learnable visual attention heatmaps; ARIAL (Mohammadshirazi et al., 2025a) adopts semantic retrieval for evidence localization; EaGERS (Hormazábal et al., 2025) localizes evidence regions through multimodal embedding similarity and majority voting; DLaVA (Mohammadshirazi et al., 2025b) leverages multimodal large language models for answer region localization; and GAP (Le et al., 2023) utilizes attention priors to compute attention weights for localization. For the trainable baseline methods, we follow the same training protocol as our approach and adopt identical backbone networks to ensure fair comparison.

**Evaluation Metrics.** Answer generation quality is evaluated using EM, F1, and ANLS metrics. Evidence localization accuracy is evaluated using the mAP@IoU metric, which computes the mean average precision over IoU thresholds ranging from 0.5 to 0.9 with a step size of 0.05, thereby providing a comprehensive evaluation under varying localization precision requirements. Additional experimental details are provided in Appendix B.

*Table 2.* Evidence localization performance on DUDE under different IoU thresholds

| Method | AP at IoU Threshold | | | | |
|---|---|---|---|---|---|
| | @0.5 | @0.6 | @0.7 | @0.8 | @0.9 |
| GAP | 66.9 | 58.7 | 47.3 | 30.2 | 12.8 |
| DLaVA | 82.5 | 75.6 | 63.4 | 43.1 | 19.2 |
| EaGERS | 68.4 | 61.2 | 56.7 | 39.5 | 17.3 |
| ARIAL | 78.3 | 70.4 | 57.8 | 39.7 | 21.6 |
| DocVXQA | 77.8 | 72.5 | 65.9 | 43.2 | 23.4 |
| **Ours** | **84.3** | **76.5** | **69.6** | **52.9** | **25.3** |

*Table 3.* Evidence localization performance on NiM-Benchmark under different IoU thresholds

| Method | AP at IoU Threshold | | | | | |
|---|---|---|---|---|---|---|
| | @0.65 | @0.7 | @0.75 | @0.8 | @0.85 | @0.9 |
| GAP | 39.7 | 34.3 | 29.5 | 24.1 | 17.8 | 11.6 |
| DLaVA | 50.6 | 45.2 | 40.3 | 34.8 | 27.5 | 19.4 |
| EaGERS | 44.2 | 38.6 | 33.8 | 28.2 | 21.1 | 14.3 |
| ARIAL | 47.8 | 42.1 | 37.2 | 31.5 | 24.3 | 16.8 |
| DocVXQA | 52.3 | 47.1 | 42.5 | 37.2 | 29.8 | 21.7 |
| **Ours** | **60.1** | **53.3** | **48.1** | **41.9** | **33.9** | **26.4** |

### 4.2. Main Results

Table 1 compares the proposed method with representative baselines on three benchmarks. Our method achieves the best performance across all evaluation metrics, demonstrating the effectiveness of the proposed framework. For the key metric mAP@IoU, our method consistently outperforms the strongest baseline, DocVXQA, on DocVQA, NiM-Benchmark, and DUDE. The improvement is particularly pronounced on NiM-Benchmark, where evidence regions occupy only a small portion of the page and the document semantics are highly dense. This result suggests that the progressive evolution induced by the Wasserstein gradient flow can effectively refine evidence boundaries, thereby alleviating the boundary ambiguity and localization instability suffered by existing methods. The answer generation metrics, including EM, F1, and ANLS, also show consistent gains across the three datasets. This can be attributed to the tight coupling between localization and generation in our framework: clearer evidence boundaries reduce contextual noise passed to the answer decoder, enabling the model to generate more faithful answers.

### 4.3. Precise Evidence Region Localization

Tables 2 and 3 compare evidence localization accuracy under different IoU thresholds on DUDE and NiM-Benchmark, respectively. As the IoU threshold becomes increasingly stringent, our method consistently demonstrates superior

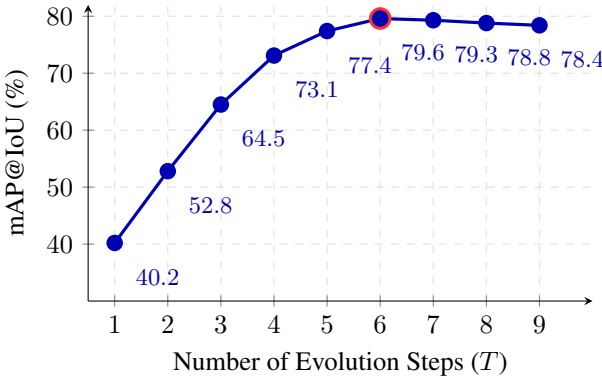

*Figure 3.* Effect of evolution steps on localization performance

performance compared with existing approaches. This result indicates that the proposed framework not only identifies the correct evidence regions but also delineates their spatial boundaries with higher precision.

These advantages are a direct consequence of the proposed progressive evolution mechanism. By gradually transporting probability mass in the distribution space along the gradient of the energy functional, the candidate evidence distribution is continuously refined and progressively converges toward the true evidence boundaries, thereby maintaining high localization confidence even under strict matching criteria. The experimental results suggest that the performance gains primarily originate from more accurate boundary delineation rather than coarse-grained region matching. Furthermore, in NiM-Benchmark, evidence regions typically occupy only a very small fraction of an entire page, making it particularly susceptible to boundary collapse and region drift. Nevertheless, our method remains robust in the high-IoU regime, further demonstrating the effectiveness of the proposed mechanism for small-object and sparse-evidence scenarios. Qualitative visualizations of the progressive evidence localization process are provided in Appendix B.3.

### 4.4. Ablation Study

We conduct a series of ablation studies on the DocVQA dataset to systematically investigate the impact of key hyperparameter choices on the effectiveness of our method.

**Effect of the Number of Evolution Steps**. Figure 3 illustrates the impact of the number of evolution steps $T$ on localization performance. When $T = 1$, the model degenerates into a one-step direct localization framework, whose performance is significantly inferior to that of multi-step evolution settings. This result indirectly validates the fundamental advantage of "progressive distribution transport" over "one-shot prediction": precise delineation of evidence boundaries cannot be achieved through a single mapping, but instead re-

*Table 4.* Impact of hyperparameters $\lambda$ and $\tau$ on performance

| $\lambda$ | $\tau$ | mAP@IoU | $\lambda$ | $\tau$ | mAP@IoU |
|------|------|------|------|------|------|
| 0.5 | 0.05 | 68.5 | 0.75 | 0.05 | 72.3 |
| 1.0 | 0.05 | 74.8 | 1.25 | 0.05 | 71.6 |
| 1.5 | 0.05 | 67.9 | 2.0 | 0.05 | 63.2 |
| 0.5 | 0.1 | 71.4 | 0.75 | 0.1 | 76.8 |
| **1.0** | **0.1** | **79.6** | 1.25 | 0.1 | 77.2 |
| 1.5 | 0.1 | 72.5 | 2.0 | 0.1 | 66.8 |

quires iterative refinement along the gradient direction in the distribution space. From $T = 2$ to $T = 4$, each additional evolution step consistently improves performance, indicating that the majority of the distribution is still migrating toward the ground-truth evidence regions during this stage. After $T = 5$, the distribution gradually approaches the target distribution, and subsequent evolution primarily focuses on boundary refinement; consequently, the performance gain naturally diminishes and reaches its peak at $T = 6$. When $T$ continues to increase, performance exhibits a slight decline. This phenomenon is consistent with the intrinsic characteristics of discretized Wasserstein gradient flow: excessive evolution steps amplify discretization errors at each step and accumulate them over time, while also potentially causing the distribution to over-oscillate around already converged regions, thereby degrading the final localization accuracy. These results suggest that increasing the number of evolution steps is not always beneficial, and that the optimal value is determined by the trade-off between refinement gains and error accumulation. Based on this balance, we choose $T = 6$ as the optimal number of evolution steps.

**Hyperparameter Analysis**. Table 4 investigates the joint effects of the entropy weight $\lambda$ and the time step size $\tau$ on localization performance. The substantial gap between the best and worst configurations indicates that both hyperparameters play a critical role in determining model performance. When the entropy weight $\lambda$ is too small, the distribution lacks sufficient regularization, resulting in unstable optimization behavior. In contrast, excessively large values of $\lambda$ cause the entropy term to dominate the energy functional, making precise identification of evidence boundaries difficult. The best trade-off is achieved at $\lambda = 1.0$, which preserves adequate regularization to stabilize the distribution while maintaining sufficient discriminability of evidence boundaries. When the time step size $\tau$ is too small, the transport magnitude at each step becomes limited, making it difficult for the distribution to converge sufficiently within a fixed number of evolution steps. Moderately increasing $\tau$ accelerates convergence without introducing numerical instability. Notably, the performance gains brought by $\tau$ are most significant around $\lambda = 1.0$, while they diminish considerably when $\lambda$ deviates from its optimal value. This observation suggests that the two hyperparameters are

not independent; instead, they jointly govern the discretized evolution dynamics of the gradient flow and collectively determine whether the discrete trajectory can stably converge to the target distribution of the continuous gradient flow. Based on these observations, we finally choose $\lambda = 1.0$ and $\tau = 0.1$ as the optimal hyperparameter configuration.

## 5. Conclusion

This paper proposes a progressive evidence localization framework based on Wasserstein gradient flow. By reformulating precise evidence localization as an optimal transport optimization problem over probability distributions and deriving an end-to-end trainable loss function based on the JKO discretization scheme, the proposed method transforms the theoretical framework into an objective that can be directly optimized by neural networks, enabling progressive coarse-to-fine evolution of evidence regions. Experimental results demonstrate that the proposed framework significantly outperforms existing approaches in both evidence localization and answer generation, while simultaneously providing an interpretable progressive reasoning process. The proposed framework offers a new perspective for achieving reliable and traceable reasoning in DocVQA systems.

## Acknowledgements

The authors would like to sincerely thank the anonymous reviewers for their valuable comments and constructive suggestions, which have improved the quality of this paper.

## Impact Statement

This paper presents work whose goal is to advance the field of machine learning. There are many potential societal consequences of our work, none of which we feel must be specifically highlighted here.

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

# A. Appendix

## A.1. Additional Definitions and Theoretical Results

Wasserstein gradient flows provide a principled framework for evolving probability distributions via the steepest descent under the Wasserstein metric.

**Definition A.1** (Wasserstein Distance). The 2-Wasserstein distance $\mathcal{W}_2 : \mathcal{P}(\Omega^*) \times \mathcal{P}(\Omega^*) \to \mathbb{R}_{\geq 0}$ is defined as

$$\mathcal{W}_2(\mu, \nu) = \left( \inf_{\pi \in \Pi(\mu, \nu)} \int_{\Omega^* \times \Omega^*} \|x - y\|^2 \, d\pi(x, y) \right)^{1/2} \tag{28}$$

where $\Pi(\mu, \nu)$ denotes the set of all coupling measures (joint distributions) whose marginals are $\mu$ and $\nu$, respectively. Intuitively, $\mathcal{W}_2(\mu, \nu)$ represents the minimum cost required to "transport" the distribution $\mu$ to $\nu$, where moving a unit mass over a distance $d$ incurs a cost of $d^2$.

**Definition A.2** (Wasserstein Gradient Flow). The Wasserstein gradient flow of the energy functional $\mathcal{E}$ in the space $(\mathcal{P}(\Omega^*), \mathcal{W}_2)$ is defined as a measure-valued curve $\{\mu_t\}_{t \geq 0}$ satisfying the following continuity equation:

$$\frac{\partial \mu_t}{\partial t} = \nabla \cdot \left( \mu_t \nabla \frac{\delta \mathcal{E}}{\delta \mu}[\mu_t] \right) \tag{29}$$

where $\nabla$ denotes the spatial gradient operator on $\Omega^*$, $\nabla \cdot$ denotes the divergence operator, and $\frac{\delta \mathcal{E}}{\delta \mu}$ denotes the first variation of the energy functional $\mathcal{E}$ with respect to the measure $\mu$.

**Lemma A.3** (Computation of the First Variation). *The first variation of the energy functional in Eq. (6) is given by*

$$\frac{\delta \mathcal{E}}{\delta \mu}(x) = V(x; Q, A) - \lambda\big(1 + \log \rho(x)\big) + 2\gamma \phi(x) \tag{30}$$

*where $\phi : \Omega^* \to \mathbb{R}$ is the Kantorovich potential associated with the Wasserstein distance, satisfying*

$$\nabla \phi(x) = x - T_{\mu \to \mu^*}(x) \tag{31}$$

*and $T_{\mu \to \mu^*}$ denotes the optimal transport map from $\mu$ to the target distribution $\mu^*$.*

**Lemma A.4** (Relationship Between the JKO Scheme and Gradient Flows). *(Jordan et al., 1998; Ambrosio et al., 2005) As the time step $\tau \to 0$, the discrete sequence $\{\mu^{(k)}\}$ generated by the JKO scheme converges to the continuous Wasserstein gradient flow $\{\mu_t\}$. Specifically, define the piecewise-constant interpolation*

$$\mu_t^\tau = \mu^{(\lfloor t/\tau \rfloor)} \tag{32}$$

*Then $\mu_t^\tau \to \mu_t$ in the $\mathcal{W}_2$ metric, with a convergence rate of $\mathcal{O}(\tau)$.*

**Proposition A.5** (Convergence of the Discrete Flow). *Let $\{\mu^{(k)}\}_{k=0}^T$ be the sequence generated by the JKO scheme. Assume that the energy functional $\mathcal{E}$ is $\lambda$-displacement convex on $\mathcal{P}(\Omega^*)$, and let $\bar{\mu}$ be the unique minimizer of $\mathcal{E}$. Then:*

*(i) Monotonic energy dissipation:*

$$\mathcal{E}[\mu^{(k+1)}] - \mathcal{E}[\mu^{(k)}] \leq -\frac{1}{2\tau} \mathcal{W}_2^2(\mu^{(k+1)}, \mu^{(k)}) \tag{33}$$

*(ii) Sublinear convergence rate:*

$$\mathcal{E}[\mu^{(T)}] - \mathcal{E}[\bar{\mu}] \leq \frac{\mathcal{W}_2^2(\mu^{(0)}, \bar{\mu})}{2\tau T} = \mathcal{O}\left(\frac{1}{T}\right) \tag{34}$$

*(iii) Linear convergence under strong convexity: If $\kappa > 0$, that is, if $\mathcal{E}$ is strongly displacement convex, then the sequence converges to $\bar{\mu}$ at an exponential rate:*

$$\mathcal{W}_2^2(\mu^{(k)}, \bar{\mu}) \leq \left(\frac{1}{1 + \kappa \tau}\right)^k \mathcal{W}_2^2(\mu^{(0)}, \bar{\mu}) \tag{35}$$

**Corollary A.6** (Distribution Error Induced by Potential Approximation). *Under the conditions of Proposition 3.9, the JKO sequence is also linearly bounded by the approximation error in terms of the Wasserstein distance:*

$$\lim_{T \to \infty} \mathcal{W}_2^2(\mu^{(T)}, \bar{\mu}) \leq \frac{4\,\epsilon_{approx}}{\kappa} \tag{36}$$

*where $\kappa = \lambda_V + 2\gamma > 0$ is the displacement convexity constant of $\mathcal{E}$, as established in Proposition 3.5.*

**Proposition A.7** (Single-Step Approximation Error of Parameterized JKO). *Let $\mu_*^{(k+1)}$ denote the exact minimizer of the k-th JKO step obtained by minimizing over the entire space $\mathcal{P}(\Omega^*)$, and let $\mu_\theta^{(k+1)}$ denote the approximate solution within the parameterized family $\mathcal{M}_\Theta = \{\mu_\theta : \theta \in \Theta\}$. Define the single-step parameterization error as*

$$\delta_{param}^{(k)} = \inf_{\theta \in \Theta} \mathcal{W}_2(\mu_\theta, \mu_*^{(k+1)}) \tag{37}$$

*Assume that the energy functional $\mathcal{E}$ is $\kappa$-displacement convex ($\kappa = \lambda_V + 2\gamma > 0$) and L-smooth. Then, the parameterized JKO solution $\mu_\theta^{(k+1)}$ satisfies*

$$\mathcal{W}_2(\mu_\theta^{(k+1)}, \mu_*^{(k+1)}) \leq C_1 \cdot \delta_{param}^{(k)} \tag{38}$$

*where $C_1 = \sqrt{\frac{1+L\tau}{1+\kappa\tau}}$ is the single-step error amplification factor.*

**Definition A.8** (Information-Theoretic Chain Rule). The information-theoretic chain rule can be expressed as

$$I(x; A \mid Q) = \underbrace{I(x; A)}_{\text{intrinsic information}} + \underbrace{\left(I(x; A \mid Q) - I(x; A)\right)}_{\text{information gain induced by the question}} \tag{39}$$

where the first term captures the intrinsic information provided by $x$ about the answer $A$, and the second term characterizes the additional information gain conditioned on the question $Q$.

**Definition A.9** (Multi-Head Attention Mechanism). For notational convenience, we define the multi-head attention mechanism $\mathcal{M}$ as a composite operator that incorporates residual connections and layer normalization. Given the query $Q \in \mathbb{R}^{n \times d}$, key $K \in \mathbb{R}^{m \times d}$, and value $V \in \mathbb{R}^{m \times d}$, the operator $\mathcal{M} : \mathbb{R}^{n \times d} \times \mathbb{R}^{m \times d} \times \mathbb{R}^{m \times d} \to \mathbb{R}^{n \times d}$ is defined as

$$\mathcal{M}(Q, K, V) = \text{LayerNorm}\big(Q + \text{MultiHead}(Q, K, V)\big) \tag{40}$$

Here, MultiHead$(\cdot)$ denotes the standard multi-head attention operation:

$$\text{MultiHead}(Q, K, V) = \text{Concat}(\text{head}_1, \ldots, \text{head}_h)\, W^O \tag{41}$$

$$\text{head}_i = \text{Attention}(QW_i^Q,\ KW_i^K,\ VW_i^V) \tag{42}$$

$$\text{Attention}(Q', K', V') = \text{Softmax}\left(\frac{Q'K'^\top}{\sqrt{d_k}}\right) V' \tag{43}$$

Here, $h$ denotes the number of attention heads, $d_k = d/h$ denotes the dimensionality of each head, $W_i^Q, W_i^K, W_i^V \in \mathbb{R}^{d \times d_k}$ are the projection matrices of the $i$-th head, $W^O \in \mathbb{R}^{d \times d}$ is the output projection matrix, and LayerNorm$(\cdot)$ denotes the layer normalization operation.

*Remark* A.10. In this definition, $\mathcal{M}$ explicitly includes residual connections and layer normalization. This convention is adopted to simplify the mathematical notation throughout the paper, allowing $\mathcal{M}$ to directly represent the corresponding module operations in the proposed framework.

**Definition A.11** (Feed-Forward Neural Network). Consistent with the convention adopted in Definition A.9, we define the feed-forward neural network $\mathcal{F}_N : \mathbb{R}^{n \times d} \to \mathbb{R}^{n \times d}$ as a composite operator incorporating residual connections and layer normalization:

$$\mathcal{F}_N(X) = \text{LayerNorm}\big(X + \text{FFN}(X)\big). \tag{44}$$

Here, FFN$(\cdot)$ denotes a two-layer fully connected feed-forward network:

$$\text{FFN}(X) = \sigma(XW_1 + b_1)\, W_2 + b_2, \tag{45}$$

where $W_1 \in \mathbb{R}^{d \times d_{ff}}$ and $W_2 \in \mathbb{R}^{d_{ff} \times d}$ are weight matrices, $b_1 \in \mathbb{R}^{d_{ff}}$ and $b_2 \in \mathbb{R}^d$ are bias vectors, and $\sigma(\cdot)$ denotes the activation function (ReLU in this work).

### A.2. Proof

### 1.Proof of Proposition 3.5

*Proof.* The proof proceeds in three steps.

**Step 1: Monotonic energy dissipation along the flow.**

Define the Lyapunov functional $\mathcal{L}(t) = \mathcal{E}[\mu_t; Q, A]$. Along the gradient flow (8), we have

$$\frac{d\mathcal{L}}{dt} = \int_{\Omega^*} \frac{\delta\mathcal{E}}{\delta\mu}[\mu_t](x)\, \partial_t \rho_t(x)\, dx \tag{46}$$

$$= \int_{\Omega^*} \frac{\delta\mathcal{E}}{\delta\mu}\, \nabla \cdot \left( \rho_t \nabla \frac{\delta\mathcal{E}}{\delta\mu} \right) dx \tag{47}$$

$$= - \int_{\Omega^*} \rho_t \left| \nabla \frac{\delta\mathcal{E}}{\delta\mu} \right|^2 dx\ \leq\ 0 \tag{48}$$

The last equality follows from integration by parts together with the no-flux boundary condition

$$\rho_t \nabla \frac{\delta\mathcal{E}}{\delta\mu} \cdot \mathbf{n}\big|_{\partial\Omega^*} = 0 \tag{49}$$

Equality holds if and only if

$$\nabla \frac{\delta\mathcal{E}}{\delta\mu} = 0 \tag{50}$$

which implies that $\mu_t$ is a critical point of $\mathcal{E}$.

**Step 2: Convergence to the critical set.**

By the energy dissipation estimate and the lower boundedness of $\mathcal{E}$, the trajectory $\{\mu_t\}_{t\geq 0}$ is relatively compact in $(\mathcal{P}_2(\Omega^*), W_2)$. By the extension of LaSalle's invariance principle to Wasserstein spaces (Ambrosio et al., 2005), $\mu_t$ converges to the critical set of $\mathcal{E}$:

$$\left\{ \mu : \nabla \frac{\delta\mathcal{E}}{\delta\mu}[\mu] = 0 \right\} \tag{51}$$

**Step 3: Uniqueness and optimality of the critical point.**

By the additivity property of displacement convexity (Villani, 2021; Ambrosio et al., 2005), the functional $\mathcal{E}$ is at least $\lambda_V$-displacement convex. Since $\gamma > 0$, $\mathcal{E}$ is strictly displacement convex. Therefore, $\mathcal{E}$ admits a unique minimizer $\bar{\mu}$ in $\mathcal{P}(\Omega^*)$. Combining this result with Step 2 yields $\mu_t \to \bar{\mu}$.

the support set $R^*$ of $\bar{\mu}$ simultaneously satisfies semantic sufficiency and spatial minimality, while remaining consistent with the target distribution $\mu^*$ through the transport term. Consequently, $R^*$ is the solution to the optimization problem (3).

*Remark.* The condition $\lambda_V > 0$ corresponds to a convexity assumption, which is equivalent to the local strong convexity of $-\log p(A \mid x, Q)$ with respect to $x$. In practical VQA scenarios, this condition typically holds only locally. Accordingly, the above result should be interpreted as establishing local uniqueness and local convergence. □

### 2.Proof of Proposition A.5

*Proof.* **(i) Monotonic energy dissipation.**

By the minimization principle of the JKO scheme (9), for any $\nu \in \mathcal{P}(\Omega^*)$, we have

$$\frac{1}{2\tau}\mathcal{W}_2^2(\mu^{(k+1)}, \mu^{(k)}) + \mathcal{E}[\mu^{(k+1)}] \leq \frac{1}{2\tau}\mathcal{W}_2^2(\nu, \mu^{(k)}) + \mathcal{E}[\nu] \tag{52}$$

Taking $\nu = \mu^{(k)}$ and using $\mathcal{W}_2^2(\mu^{(k)}, \mu^{(k)}) = 0$, we immediately obtain

$$\mathcal{E}[\mu^{(k+1)}] - \mathcal{E}[\mu^{(k)}] \leq -\frac{1}{2\tau}\mathcal{W}_2^2(\mu^{(k+1)}, \mu^{(k)}) \leq 0 \tag{53}$$

Thus, the energy decreases monotonically along the discrete sequence.

**(ii) Sublinear convergence rate.**

Taking $\nu = \bar{\mu}$ in (52) gives

$$\mathcal{E}[\mu^{(k+1)}] - \mathcal{E}[\bar{\mu}] \leq \frac{1}{2\tau}\left(\mathcal{W}_2^2(\bar{\mu}, \mu^{(k)}) - \mathcal{W}_2^2(\mu^{(k+1)}, \mu^{(k)})\right) \tag{54}$$

By the geometric inequality for JKO minimizers in Wasserstein space (Ambrosio et al., 2005),

$$\mathcal{W}_2^2(\bar{\mu}, \mu^{(k)}) \geq \mathcal{W}_2^2(\bar{\mu}, \mu^{(k+1)}) + \mathcal{W}_2^2(\mu^{(k+1)}, \mu^{(k)}). \tag{55}$$

Substituting this inequality and rearranging terms yield

$$\mathcal{E}[\mu^{(k+1)}] - \mathcal{E}[\bar{\mu}] \leq \frac{1}{2\tau}\left(\mathcal{W}_2^2(\bar{\mu}, \mu^{(k)}) - \mathcal{W}_2^2(\bar{\mu}, \mu^{(k+1)})\right) \tag{56}$$

Summing over $k = 0, 1, \ldots, T-1$, the right-hand side telescopes:

$$\sum_{k=0}^{T-1}\left(\mathcal{E}[\mu^{(k+1)}] - \mathcal{E}[\bar{\mu}]\right) \leq \frac{1}{2\tau}\left(\mathcal{W}_2^2(\bar{\mu}, \mu^{(0)}) - \mathcal{W}_2^2(\bar{\mu}, \mu^{(T)})\right) \leq \frac{\mathcal{W}_2^2(\mu^{(0)}, \bar{\mu})}{2\tau} \tag{57}$$

By the monotonic energy dissipation established in (i), $\mathcal{E}[\mu^{(T)}] \leq \mathcal{E}[\mu^{(k+1)}]$ for all $k = 0, 1, \ldots, T-1$. Therefore,

$$T \cdot \left(\mathcal{E}[\mu^{(T)}] - \mathcal{E}[\bar{\mu}]\right) \leq \sum_{k=0}^{T-1}\left(\mathcal{E}[\mu^{(k+1)}] - \mathcal{E}[\bar{\mu}]\right) \leq \frac{\mathcal{W}_2^2(\mu^{(0)}, \bar{\mu})}{2\tau} \tag{58}$$

Hence,

$$\mathcal{E}[\mu^{(T)}] - \mathcal{E}[\bar{\mu}] \leq \frac{\mathcal{W}_2^2(\mu^{(0)}, \bar{\mu})}{2\tau T} = \mathcal{O}\left(\frac{1}{T}\right) \tag{59}$$

**(iii) Linear convergence under strong convexity.**

By the $\kappa$-displacement convexity of $\mathcal{E}$ with $\kappa > 0$, for any $\mu \in \mathcal{P}(\Omega^*)$ (Ambrosio et al., 2005), we have

$$\mathcal{E}[\mu] - \mathcal{E}[\bar{\mu}] \geq \frac{\kappa}{2}\mathcal{W}_2^2(\mu, \bar{\mu}) \tag{60}$$

Applying (60) to the left-hand side of (56) gives

$$\frac{\kappa}{2}\mathcal{W}_2^2(\mu^{(k+1)}, \bar{\mu}) \leq \mathcal{E}[\mu^{(k+1)}] - \mathcal{E}[\bar{\mu}] \leq \frac{1}{2\tau}\left(\mathcal{W}_2^2(\bar{\mu}, \mu^{(k)}) - \mathcal{W}_2^2(\bar{\mu}, \mu^{(k+1)})\right) \tag{61}$$

Multiplying both sides by $2\tau$ and rearranging, we obtain

$$\kappa\tau \cdot \mathcal{W}_2^2(\mu^{(k+1)}, \bar{\mu}) \leq \mathcal{W}_2^2(\bar{\mu}, \mu^{(k)}) - \mathcal{W}_2^2(\bar{\mu}, \mu^{(k+1)}) \tag{62}$$

Collecting like terms yields

$$(1 + \kappa\tau)\mathcal{W}_2^2(\mu^{(k+1)}, \bar{\mu}) \leq \mathcal{W}_2^2(\mu^{(k)}, \bar{\mu}) \tag{63}$$

Equivalently,

$$\mathcal{W}_2^2(\mu^{(k+1)}, \bar{\mu}) \leq \frac{1}{1+\kappa\tau} \mathcal{W}_2^2(\mu^{(k)}, \bar{\mu}) \tag{64}$$

Iterating this inequality for $k$ steps gives

$$\mathcal{W}_2^2(\mu^{(k)}, \bar{\mu}) \leq \left(\frac{1}{1+\kappa\tau}\right)^k \mathcal{W}_2^2(\mu^{(0)}, \bar{\mu}) \tag{65}$$

Since $\kappa > 0$ and $\tau > 0$, we have $\frac{1}{1+\kappa\tau} < 1$. Therefore, the sequence $\{\mu^{(k)}\}$ converges exponentially to the unique minimizer $\bar{\mu}$. $\square$

### 3.Proof of Proposition 3.8

*Proof.* By the JKO scheme (9) and the energy functional (6), the minimization objective at the $k$-th step is

$$\mu^{(k+1)} = \arg\min_\mu \left\{ \int V(x; Q, A) \, d\mu(x) - \lambda \int \rho \log \rho \, dx + \gamma \mathcal{W}_2^2(\mu, \mu^*) + \frac{1}{2\tau} \mathcal{W}_2^2(\mu, \mu^{(k)}) \right\} \tag{66}$$

Under the neural parameterization in (10), each term admits a differentiable expression with respect to $\theta$.

**Potential energy term.** Since $f_\theta(x, Q) \approx V(x; Q, A)$, and the candidate distribution $\mu^{(k+1)}$ is represented by $\mu_\theta$, we have

$$\int V(x; Q, A) \, d\mu^{(k+1)}(x) \approx \mathbb{E}_{x \sim \mu_\theta}[f_\theta(x, Q)] \tag{67}$$

This gives $\mathcal{L}_{\text{pot}}(\theta)$.

**Entropy term.** Since $\rho_\theta = \exp(-f_\theta)/Z_\theta$, we have $\log \rho_\theta(x) = -f_\theta(x, Q) - \log Z_\theta(Q)$. Therefore,

$$\int \rho_\theta(x) \log \rho_\theta(x) \, dx = \int \rho_\theta(x) \left(-f_\theta(x, Q) - \log Z_\theta(Q)\right) dx \tag{68}$$

$$= -\mathbb{E}_{x \sim \mu_\theta}[f_\theta(x, Q)] - \log Z_\theta(Q) \tag{69}$$

Thus, the entropy loss is

$$\mathcal{L}_{\text{ent}}(\theta) = -\int \rho_\theta \log \rho_\theta \, dx = \mathbb{E}_{x \sim \mu_\theta}[f_\theta(x, Q)] + \log Z_\theta(Q) \tag{70}$$

**Target-distribution control term.** We replace $\mathcal{W}_2^2(\mu_\theta, \mu^*)$ with its differentiable Sinkhorn approximation $\hat{\mathcal{W}}_2^2$:

$$\gamma \mathcal{W}_2^2(\mu_\theta, \mu^*) \approx \gamma \hat{\mathcal{W}}_2^2(\mu_\theta, \mu^*) = \mathcal{L}_{\text{prior}}(\theta) \tag{71}$$

**JKO transport penalty.** Similarly,

$$\frac{1}{2\tau} \mathcal{W}_2^2(\mu_\theta, \mu^{(k)}) \approx \frac{1}{2\tau} \hat{\mathcal{W}}_2^2(\mu_\theta, \mu^{(k)}) = \mathcal{L}_{\text{trans}}^{(k)}(\theta) \tag{72}$$

Combining the four terms yields the loss function in (11). $\square$

### 4.Proof of Proposition 3.9

*Proof.* By the definition of the energy functional in (6), $\tilde{\mathcal{E}}$ and $\mathcal{E}$ differ only in the potential energy term. Thus, for any $\mu \in \mathcal{P}(\Omega^*)$, we have

$$|\tilde{\mathcal{E}}[\mu] - \mathcal{E}[\mu]| = \left| \int_{\Omega^*} \left(V_\theta(x) - V^*(x)\right) d\mu(x) \right|$$

$$\leq \int_{\Omega^*} |V_\theta(x) - V^*(x)| \, d\mu(x)$$

$$\leq \epsilon_{\text{approx}} \cdot \mu(\Omega^*) = \epsilon_{\text{approx}} \tag{73}$$

By the triangle inequality and optimality, we obtain

$$\mathcal{E}[\tilde{\mu}] - \mathcal{E}[\bar{\mu}] = \left(\mathcal{E}[\tilde{\mu}] - \tilde{\mathcal{E}}[\tilde{\mu}]\right) + \left(\tilde{\mathcal{E}}[\tilde{\mu}] - \tilde{\mathcal{E}}[\bar{\mu}]\right) + \left(\tilde{\mathcal{E}}[\bar{\mu}] - \mathcal{E}[\bar{\mu}]\right) \leq 2\epsilon_{\text{approx}}$$

By Proposition A.5(iii), the JKO sequence $\{\mu^{(k)}\}$ associated with the approximate energy functional $\tilde{\mathcal{E}}$ converges exponentially to $\tilde{\mu}$, since $\tilde{\mathcal{E}}$ has the same displacement convexity constant as $\mathcal{E}$. By the lower semicontinuity of $\mathcal{E}$ on $(\mathcal{P}(\Omega^*), \mathcal{W}_2)$ (Ambrosio et al., 2005), we have

$$\lim_{T \to \infty} \mathcal{E}[\mu^{(T)}] = \mathcal{E}[\tilde{\mu}] \tag{74}$$

Therefore,

$$\lim_{T \to \infty} \mathcal{E}[\mu^{(T)}] - \mathcal{E}[\bar{\mu}] = \mathcal{E}[\tilde{\mu}] - \mathcal{E}[\bar{\mu}] \leq 2\,\epsilon_{\text{approx}} \tag{75}$$

$\square$

## 5.Proof of Corollary A.6

*Proof.* By the $\kappa$-displacement convexity of $\mathcal{E}$ established in Proposition 3.5, for any $\mu \in \mathcal{P}(\Omega^*)$, $\mathcal{E}$ satisfies the following quadratic growth condition (Ambrosio et al., 2005):

$$\mathcal{E}[\mu] - \mathcal{E}[\bar{\mu}] \geq \frac{\kappa}{2}\,\mathcal{W}_2^2(\mu, \bar{\mu}) \tag{76}$$

Taking $\mu = \tilde{\mu}$ and combining this inequality with in Proposition 3.9, we obtain

$$\frac{\kappa}{2}\,\mathcal{W}_2^2(\tilde{\mu}, \bar{\mu}) \leq \mathcal{E}[\tilde{\mu}] - \mathcal{E}[\bar{\mu}] \leq 2\,\epsilon_{\text{approx}} \tag{77}$$

Thus, $\mathcal{W}_2^2(\tilde{\mu}, \bar{\mu}) \leq \frac{4\,\epsilon_{\text{approx}}}{\kappa}$.

By Proposition 3.9, $\mu^{(T)} \to \tilde{\mu}$ under the $\mathcal{W}_2$ metric. By the continuity of the $\mathcal{W}_2$ metric,

$$\lim_{T \to \infty} \mathcal{W}_2^2(\mu^{(T)}, \bar{\mu}) = \mathcal{W}_2^2(\tilde{\mu}, \bar{\mu}) \leq \frac{4\,\epsilon_{\text{approx}}}{\kappa} \tag{78}$$

$\square$

## 6.Proof of Proposition A.7

*Proof.* Define the JKO objective functional as

$$\mathcal{J}^{(k)}[\mu] := \frac{1}{2\tau}\mathcal{W}_2^2(\mu, \mu^{(k)}) + \mathcal{E}[\mu] \tag{79}$$

Since $\mathcal{W}_2^2(\cdot, \mu^{(k)})$ is $\frac{1}{\tau}$-displacement convex and $\mathcal{E}$ is $\kappa$-displacement convex, $\mathcal{J}^{(k)}$ is $\left(\frac{1}{\tau} + \kappa\right)$-displacement convex (Ambrosio et al., 2005).

By the optimality of $\mathcal{J}^{(k)}$ at $\mu_*^{(k+1)}$ and its displacement convexity, taking $\mu = \mu_\theta^{(k+1)}$ yields

$$\mathcal{J}^{(k)}[\mu_\theta^{(k+1)}] - \mathcal{J}^{(k)}[\mu_*^{(k+1)}] \geq \frac{1 + \kappa\tau}{2\tau}\,\mathcal{W}_2^2(\mu_\theta^{(k+1)}, \mu_*^{(k+1)}) \tag{80}$$

Let $\tilde{\mu}_\theta \in \mathcal{M}_\Theta$ be the element in $\mathcal{M}_\Theta$ closest to $\mu_*^{(k+1)}$, namely $\mathcal{W}_2(\tilde{\mu}_\theta, \mu_*^{(k+1)}) = \delta_{\text{param}}^{(k)}$ By the $L$-smoothness of $\mathcal{J}^{(k)}$ (Ambrosio et al., 2005), we have

$$\mathcal{J}^{(k)}[\tilde{\mu}_\theta] - \mathcal{J}^{(k)}[\mu_*^{(k+1)}] \leq \frac{1 + L\tau}{2\tau}\left(\delta_{\text{param}}^{(k)}\right)^2 \tag{81}$$

Since $\mu_\theta^{(k+1)} = \arg\min_{\mu \in \mathcal{M}_\Theta} \mathcal{J}^{(k)}[\mu]$, we obtain

$$\mathcal{J}^{(k)}[\mu_\theta^{(k+1)}] \leq \mathcal{J}^{(k)}[\tilde{\mu}_\theta] \tag{82}$$

Combining (80), (81), and (82) gives

$$\frac{1 + \kappa\tau}{2\tau} \mathcal{W}_2^2(\mu_\theta^{(k+1)}, \mu_*^{(k+1)}) \leq \frac{1 + L\tau}{2\tau} \left(\delta_{\text{param}}^{(k)}\right)^2 \tag{83}$$

Taking the square root of both sides yields (38). $\qquad\square$

## 7.Proof of Proposition 3.10

*Proof.* Let $e_k := \mathcal{W}_2(\mu_\theta^{(k)}, \mu_*^{(k)})$ denote the deviation at the $k$-th step.

By Proposition A.5(iii), the exact JKO operator $\mathcal{T} : \mu \mapsto \arg\min_\nu \left\{\frac{1}{2\tau}\mathcal{W}_2^2(\nu, \mu) + \mathcal{E}[\nu]\right\}$ is $q$-contractive under $\kappa$-displacement convexity (Ambrosio et al., 2005):

$$\mathcal{W}_2(\mathcal{T}\mu, \mathcal{T}\nu) \leq q\,\mathcal{W}_2(\mu, \nu), \quad q = \frac{1}{\sqrt{1 + \kappa\tau}} \tag{84}$$

By Proposition A.7, we view $\mu_\theta^{(k+1)}$ as a perturbation of $\mathcal{T}\mu_\theta^{(k)}$. Then,

$$\begin{aligned} e_{k+1} &= \mathcal{W}_2(\mu_\theta^{(k+1)}, \mathcal{T}\mu_*^{(k)}) \\ &\leq \mathcal{W}_2(\mu_\theta^{(k+1)}, \mathcal{T}\mu_\theta^{(k)}) + \mathcal{W}_2(\mathcal{T}\mu_\theta^{(k)}, \mathcal{T}\mu_*^{(k)}) \end{aligned} \tag{85}$$

This yields the recursion

$$e_{k+1} \leq C_1\,\delta_{\text{param}} + q\,e_k. \tag{86}$$

Since $e_0 = 0$, iterating (86) gives

$$e_T \leq C_1\,\delta_{\text{param}} \sum_{j=0}^{T-1} q^j = C_1\,\delta_{\text{param}} \cdot \frac{1 - q^T}{1 - q} \tag{87}$$

This proves (13).

Since $q < 1$, we further obtain

$$\limsup_{T \to \infty} e_T \leq \frac{C_1\,\delta_{\text{param}}}{1 - q} \tag{88}$$

$$\square$$

## 8.Proof of Corollary 3.11

*Proof.* By the triangle inequality, we have

$$\mathcal{W}_2(\mu_\theta^{(T)}, \bar{\mu}) \leq \mathcal{W}_2(\mu_\theta^{(T)}, \mu_*^{(T)}) + \mathcal{W}_2(\mu_*^{(T)}, \bar{\mu}) \tag{89}$$

For the first term, Proposition 3.10 gives the bound $\frac{C_1}{1-q}\delta_{\text{param}}$.

For the second term, note that $\mu_*^{(T)}$ is the exact JKO iterate obtained using the approximate energy $\tilde{\mathcal{E}}$, namely the energy induced by $V_\theta$, and thus converges to $\tilde{\mu}$. By Corollary A.6,

$$\lim_{T \to \infty} \mathcal{W}_2(\mu_*^{(T)}, \bar{\mu}) = \mathcal{W}_2(\tilde{\mu}, \bar{\mu}) \leq \sqrt{\frac{4\epsilon_{\text{approx}}}{\kappa}} \tag{90}$$

Taking $\limsup_{T \to \infty}$ yields (15). $\qquad\square$

# B. Detailed Experimental Settings

## B.1. Detailed Dataset Description

NiM-Benchmark is a benchmark designed for fine-grained document visual question answering, aiming to evaluate a model's ability to localize and reason over "small-target information" in complex documents, where the target region occupies less than 5% of the image area. The benchmark contains 2,970 images and 1,180 question-answer pairs, covering six categories of real-world documents, including academic papers, newspapers, magazines, restaurant menus, webpage screenshots, and lecture slide screenshots. It further incorporates diverse question types, such as Inline and Boolean questions, to systematically evaluate fine-grained visual understanding capabilities.

DUDE is a large-scale benchmark dataset for real-world document visual question answering. It contains 5,019 visually rich English documents spanning multiple industries, domains, sources, and time periods from 1860 to 2022. The documents are multi-page, with an average length of 5.72 pages, and are paired with 41,541 highly diverse question-answer pairs. The benchmark further supports complex reasoning capabilities, including multi-hop reasoning, layout navigation, and arithmetic, comparison, and counting operations.

DocVQA is a large-scale benchmark dataset for document image visual question answering. It contains 12,767 document images sourced from the UCSF industrial document archive, covering five major industries, including tobacco, food, pharmaceuticals, fossil fuels, and chemicals. The dataset spans a wide range of document types, such as letters, forms, reports, memoranda, and invoices, covering the period from 1900 to 2018. In addition, it includes 50,000 human-annotated extractive question-answer pairs. Based on the evidence required to answer each question, the questions are categorized into nine types: handwritten, form, layout, table/list, running text, photograph, figure, yes/no, and other.

## B.2. Detailed Evaluation Metrics

**Exact Match (EM)**   The Exact Match (EM) metric measures the proportion of predictions that exactly match the ground-truth answers after normalization. It is defined as:

$$\text{EM} = \frac{1}{|Q|} \sum_{i=1}^{|Q|} \mathbb{I}\left[\text{normalize}(\hat{a}_i) = \text{normalize}(a_i)\right] \tag{91}$$

where $|Q|$ denotes the total number of evaluation questions, $\hat{a}_i$ and $a_i$ denote the predicted and ground-truth answers, respectively, and $\mathbb{I}[\cdot]$ denotes the indicator function. The normalization function $\text{normalize}(\cdot)$ sequentially performs the following operations: (1) converting all characters to lowercase; (2) removing punctuation marks; (3) removing English articles; and (4) collapsing consecutive whitespace characters. Since EM requires exact matching, it directly reflects the model's ability to generate precise answers, although it is sensitive to variations in answer phrasing.

**F1 Score**   The F1 Score evaluates partial matching quality by computing the harmonic mean of token-level precision and recall between the predicted answer and the ground-truth answer. Given the token sets $\text{Tokens}(\hat{a})$ and $\text{Tokens}(a)$ for the predicted and ground-truth answers, respectively, precision and recall are defined as:

$$\text{Precision} = \frac{|\text{Tokens}(\hat{a}) \cap \text{Tokens}(a)|}{|\text{Tokens}(\hat{a})|}, \quad \text{Recall} = \frac{|\text{Tokens}(\hat{a}) \cap \text{Tokens}(a)|}{|\text{Tokens}(a)|} \tag{92}$$

The F1 score is then computed as:

$$\text{F1} = \frac{2 \cdot \text{Precision} \cdot \text{Recall}}{\text{Precision} + \text{Recall}} \tag{93}$$

Compared with EM, the F1 Score assigns partial credit when the predicted answer partially overlaps with the ground-truth answer at the token level, thereby providing a more comprehensive assessment of semantic coverage.

**Average Normalized Levenshtein Similarity (ANLS)**   ANLS is the standard evaluation metric for document visual question answering tasks and is based on the normalized Levenshtein edit distance. Its overall formulation is given by:

$$\text{ANLS} = \frac{1}{|Q|} \sum_{i=1}^{|Q|} \max_{a_{i,j} \in A_i} s\left(\hat{a}_i, a_{i,j}\right) \tag{94}$$

where $A_i = \{a_{i,1}, a_{i,2}, \ldots, a_{i,N_i}\}$ denotes the set of all valid reference answers for the $i$-th question. The normalized Levenshtein similarity is defined using a threshold $\tau$:

$$s(\hat{a}, a) = \begin{cases} 1 - \mathrm{NL}(\hat{a}, a), & \text{if } \mathrm{NL}(\hat{a}, a) < \tau \\ 0, & \text{if } \mathrm{NL}(\hat{a}, a) \geq \tau \end{cases} \tag{95}$$

where

$$\mathrm{NL}(\hat{a}, a) = \frac{\mathrm{Lev}(\hat{a}, a)}{\max(|\hat{a}|, |a|)} \tag{96}$$

denotes the normalized edit distance, and $\mathrm{Lev}(\cdot, \cdot)$ denotes the Levenshtein edit distance. The threshold $\tau$ is set to $0.5$.

**Intersection over Union (IoU)**  Intersection over Union is the fundamental metric used to measure the spatial overlap between the predicted region and the ground-truth region. Given a predicted region $R_{\text{pred}}$ and a ground-truth region $R_{\text{gt}}$, IoU is defined as the ratio between the intersection area and the union area of the two regions:

$$\mathrm{IoU}(R_{\text{pred}}, R_{\text{gt}}) = \frac{\mathrm{Area}(R_{\text{pred}} \cap R_{\text{gt}})}{\mathrm{Area}(R_{\text{pred}} \cup R_{\text{gt}})} = \frac{\mathrm{Area}(R_{\text{pred}} \cap R_{\text{gt}})}{\mathrm{Area}(R_{\text{pred}}) + \mathrm{Area}(R_{\text{gt}}) - \mathrm{Area}(R_{\text{pred}} \cap R_{\text{gt}})} \tag{97}$$

where $\mathrm{Area}(\cdot)$ denotes the pixel area covered by a region. IoU ranges from $0$ to $1$: an IoU value of $0$ indicates no overlap between the two regions, whereas an IoU value of $1$ indicates perfect overlap. Larger IoU values correspond to higher spatial consistency between the predicted and ground-truth regions.

**Average Precision (AP) at IoU Threshold**  Average Precision (AP) under a specified IoU threshold $\theta$ jointly evaluates precision and recall for localization results, thereby reflecting the localization accuracy across different recall levels. During evaluation, all predicted regions are first sorted in descending order according to their confidence scores. For the $k$-th prediction, if it satisfies $\mathrm{IoU} \geq \theta$ with an unmatched ground-truth region, it is regarded as a true positive (TP); otherwise, it is considered a false positive (FP). The cumulative precision and recall for the top-$k$ predictions are then defined as

$$P(k) = \frac{\mathrm{TP}(k)}{k}, \qquad R(k) = \frac{\mathrm{TP}(k)}{N_{\text{gt}}} \tag{98}$$

where $N_{\text{gt}}$ denotes the number of ground-truth regions. AP is finally defined as the area under the precision–recall curve:

$$\mathrm{AP@}\theta = \sum_{k=1}^{N} P(k) \cdot \Delta R(k) \tag{99}$$

**Mean Average Precision over IoU Thresholds (mAP@IoU)**  AP computed under a single IoU threshold only reflects localization performance under a specific matching strictness and cannot comprehensively characterize model behavior under varying precision requirements. To address this limitation, we adopt the mAP@IoU metric, which averages AP over multiple IoU thresholds:

$$\mathrm{mAP@IoU} = \frac{1}{|\Theta|} \sum_{\theta \in \Theta} \mathrm{AP@}\theta \tag{100}$$

where $\Theta$ denotes the predefined set of IoU thresholds, and $|\Theta|$ denotes the number of thresholds. In this work, we adopt $\Theta = \{0.50, 0.55, 0.60, \ldots, 0.90\}$ which uniformly samples the interval $[0.50, 0.9]$ with a step size of $0.05$, resulting in a total of 9 thresholds. By averaging AP across multiple thresholds, mAP@IoU provides a comprehensive assessment of both coarse localization and fine-grained localization performance, offering greater robustness and discriminative power.

**Additional Implementation Details**  . All experiments are conducted on six NVIDIA GeForce RTX 4090 GPUs and implemented using PyTorch 2.7.1. The detailed training settings are as follows:

*Table 5.* Results on DocVQA under different choices of the total number of experts $N$ and Top-$K$ routing.

| $N$ | $K$ | EM (%) | F1 (%) | ANLS (%) | Inference Time (ms) |
|---|---|---|---|---|---|
| 4 | 1 | 35.8 | 46.5 | 52.2 | 146 |
| 4 | 2 | **43.2** | **55.8** | **62.7** | 158 |
| 4 | 3 | 40.2 | 51.3 | 57.9 | 177 |
| 4 | 4 | 37.9 | 48.7 | 55.2 | 190 |
| 6 | 1 | 33.3 | 44.2 | 50.4 | 159 |
| 6 | 2 | 39.8 | 51.6 | 58.4 | 174 |
| 6 | 3 | 41.3 | 53.8 | 60.2 | 192 |
| 6 | 4 | 38.9 | 50.5 | 56.9 | 220 |
| 6 | 5 | 36.5 | 47.8 | 54.2 | 241 |
| 6 | 6 | 34.7 | 45.4 | 51.7 | 257 |
| 8 | 1 | 31.5 | 42.4 | 48.7 | 161 |
| 8 | 2 | 37.7 | 49.2 | 56.2 | 182 |
| 8 | 3 | 39.5 | 51.2 | 58.1 | 209 |
| 8 | 4 | 38.2 | 49.9 | 56.5 | 243 |
| 8 | 5 | 36.3 | 47.7 | 54.4 | 264 |
| 8 | 6 | 34.9 | 46.1 | 52.7 | 281 |
| 8 | 7 | 33.4 | 44.6 | 51.1 | 297 |
| 8 | 8 | 32.2 | 43.3 | 49.8 | 312 |

**Optimization Settings**   We employ the Adam optimizer with parameters $\beta_1 = 0.9$, $\beta_2 = 0.999$, and $\epsilon = 10^{-8}$. The initial learning rate is set to $1.0 \times 10^{-4}$, and the weight decay is set to $1.0 \times 10^{-5}$. A cosine annealing schedule is adopted to dynamically adjust the learning rate according to:

$$\eta_t = \eta_{\min} + \frac{1}{2}(\eta_{\max} - \eta_{\min})\left(1 + \cos\left(\frac{t}{T}\pi\right)\right) \tag{101}$$

where $\eta_{\max} = 1.0 \times 10^{-4}$, $\eta_{\min} = 1.0 \times 10^{-6}$, $t$ denotes the current training epoch, and $T = 30$ denotes the total number of epochs for cosine decay. Gradient clipping with a maximum norm of $1.0$ is applied to stabilize training.

**Encoder Selection**   The image encoder adopts the visual model of the pretrained ColQwen3 model (Huang & Tan, 2025) to encode each document page, while the text encoder utilizes the Transformer blocks of the language model in the pretrained ColQwen3 model to encode the input question.

**Training Strategy**   The batch size is set to $16$, and gradient accumulation is performed over $2$ steps, resulting in an effective batch size of $32$. The model is trained for up to $100$ epochs, with early stopping applied according to validation performance. During training, mixed-precision training with BF16 is employed to reduce memory consumption and accelerate computation.

The key hyperparameter settings are summarized as follows. The evolution step number $T$ is set to 6, the negative entropy regularization weight $\lambda$ is set to 1.0, and the time step size $\tau$ is set to 0.1. The loss weights $\beta$, $\gamma$, and $\eta$ are all set to 1. In the multi-expert decoder, the number of experts is set to 4, while the Top-$K$ expert selection parameter is set to 2. The performance under different expert configurations is reported in Table 5.

In addition, Table 6 compares the impact of four evolution loss weighting strategies on localization performance. The uniform weighting strategy ($\alpha_k = 1.0$) treats all evolution steps equally and achieves only 73.2 mAP@IoU. By assigning

*Table 6.* Comparison of evolution loss weighting strategies

| Strategy | mAP@IoU (%) |
|---|---|
| $\alpha_k = 1.0$ | 73.2 |
| $\alpha_k = \frac{k+1}{T}$ | 75.8 |
| $\alpha_k = e^{k/T}$ | 77.4 |
| LW-$\alpha_k$ | **79.6** |

larger weights to later evolution stages, the linearly increasing strategy ($\alpha_k = (k+1)/T$) and the exponentially increasing strategy ($\alpha_k = e^{k/T}$) improve the performance to 75.8 and 77.4, respectively, indicating that fine-grained localization in the later evolution stages plays a more critical role in the final prediction. However, these manually designed weighting schemes cannot adaptively adjust to varying sample complexities. When $\{\alpha_k\}_{k=0}^{T-1}$ are treated as learnable parameters, the mAP@IoU further improves to 79.6, achieving a gain of 6.4 points over the uniform weighting strategy. This result validates the effectiveness of adaptive weight learning for progressive evolution.

## B.3. Visualization

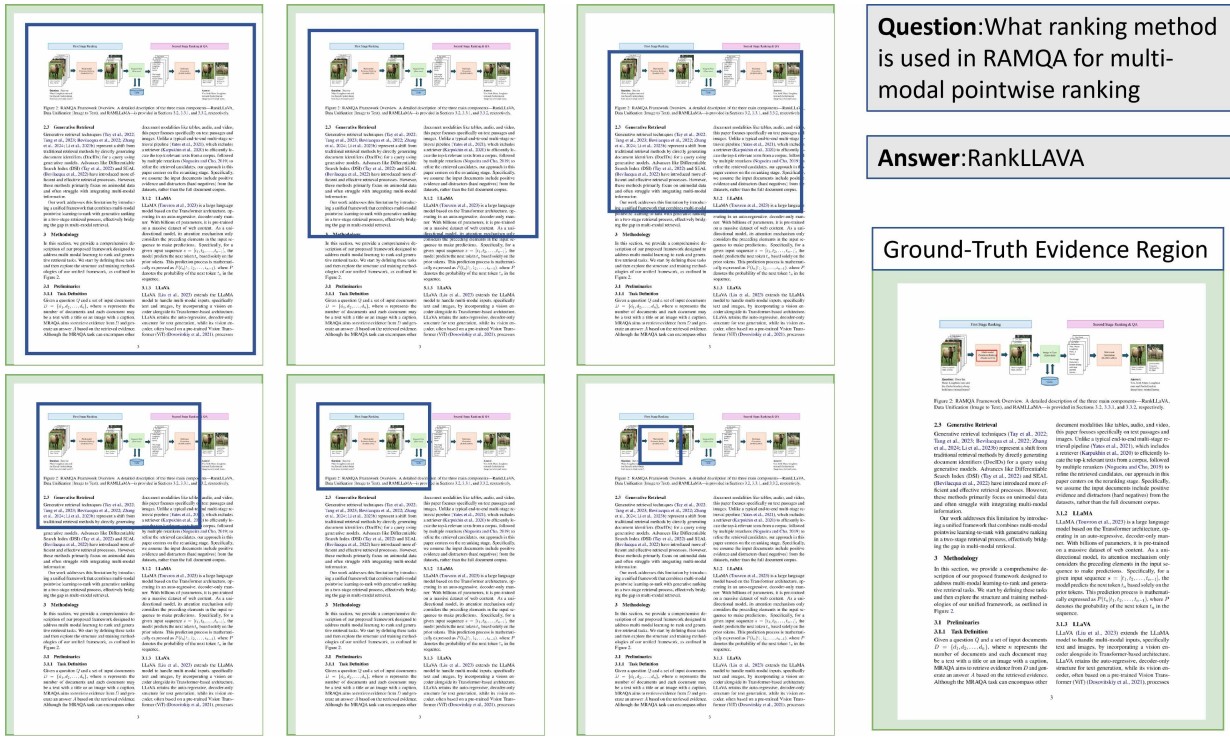

*Figure 4.* Progressive Evidence Region Localization Example

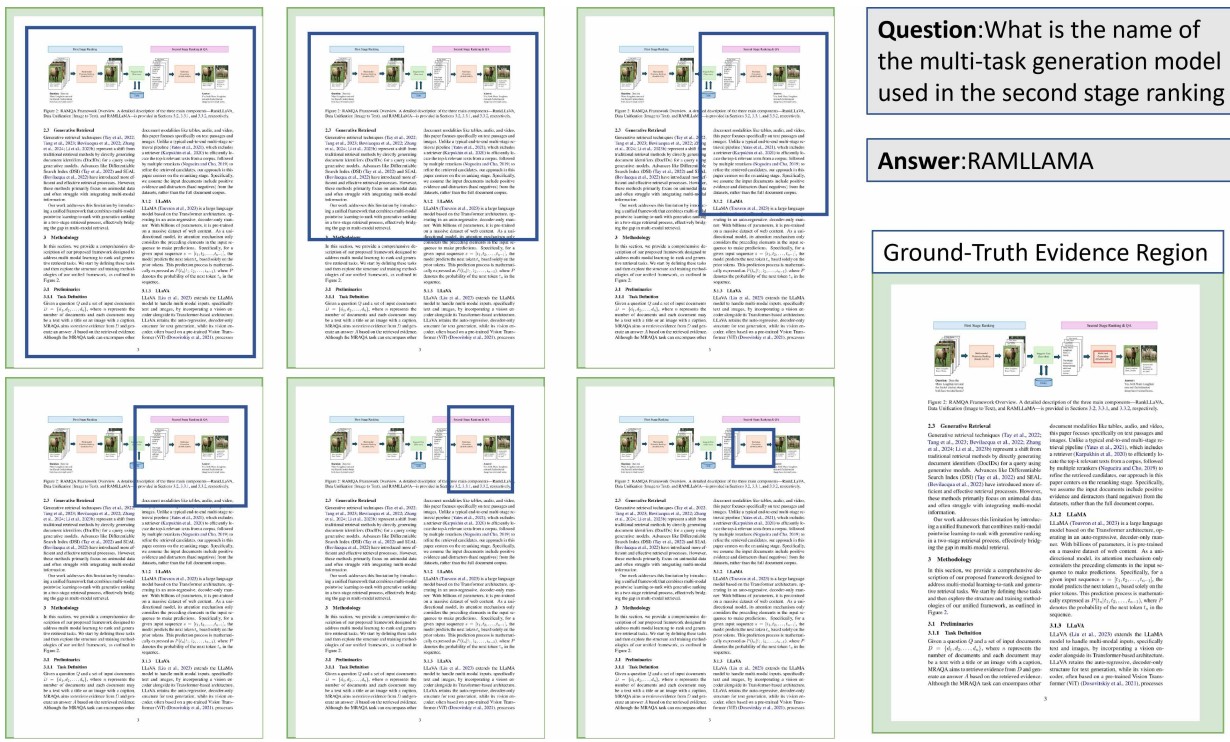

*Figure 5.* Progressive Evidence Region Localization Example

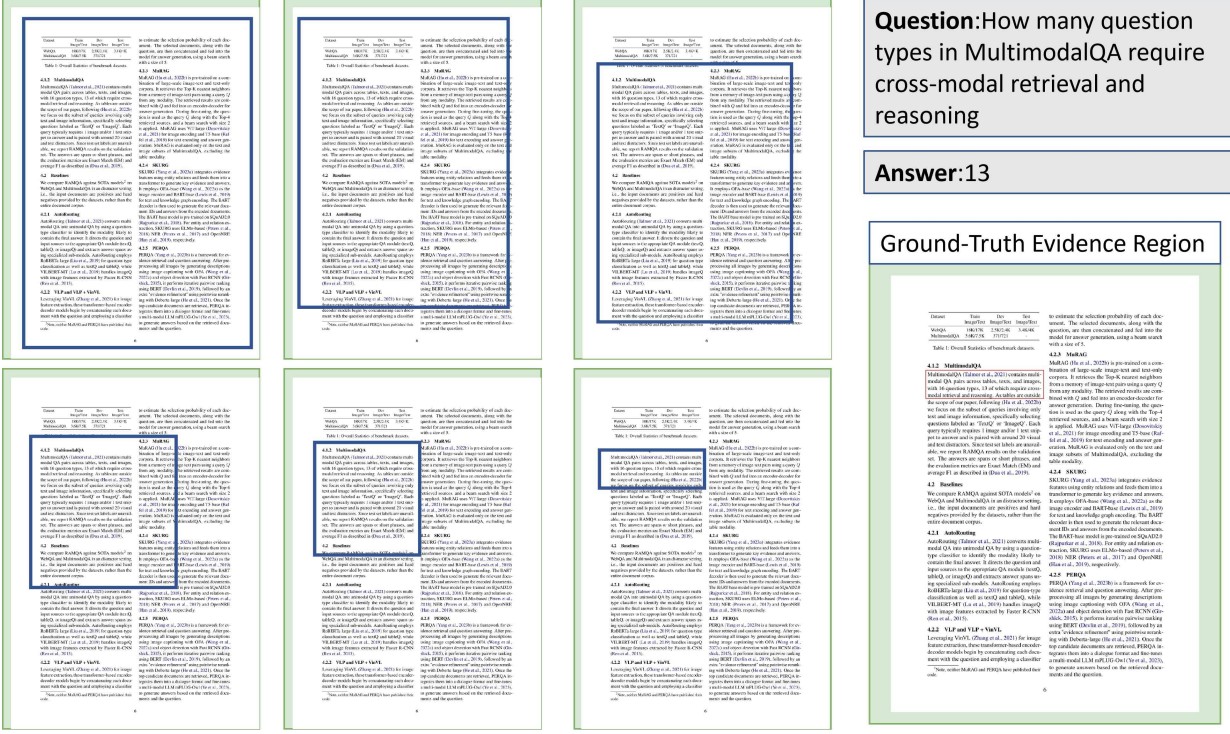

*Figure 6.* Progressive Evidence Region Localization Example

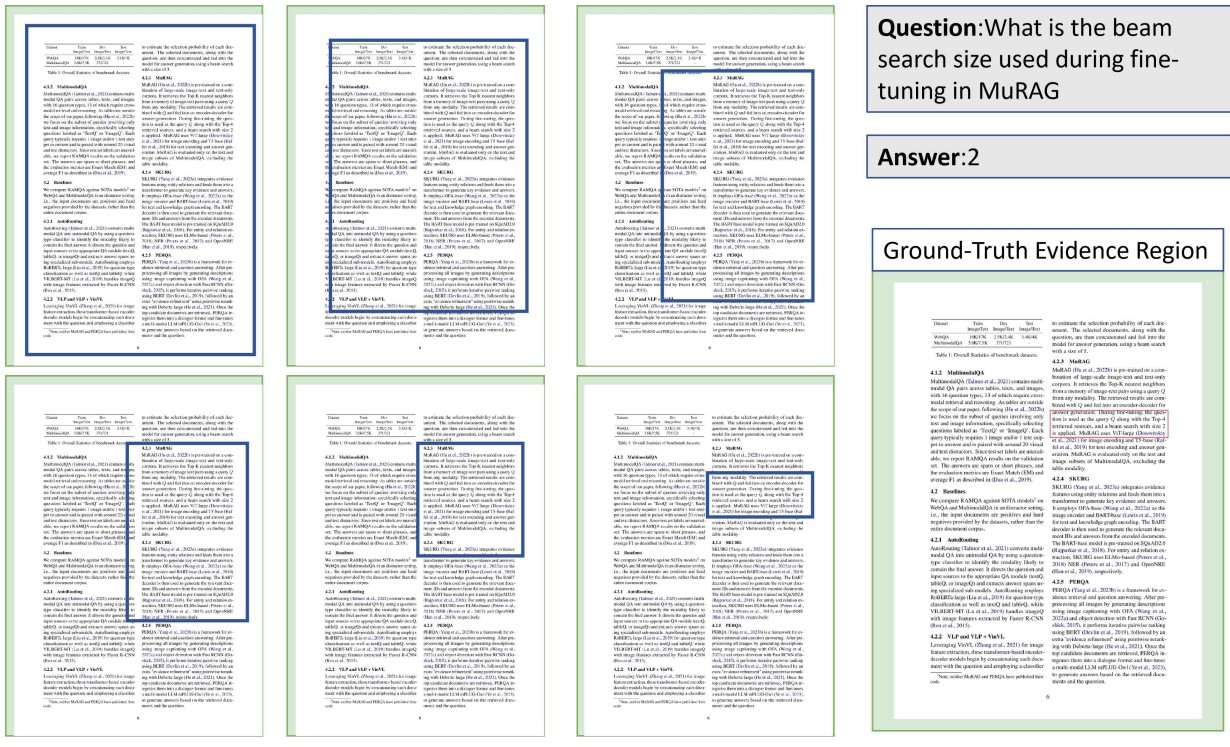

*Figure 7.* Progressive Evidence Region Localization Example

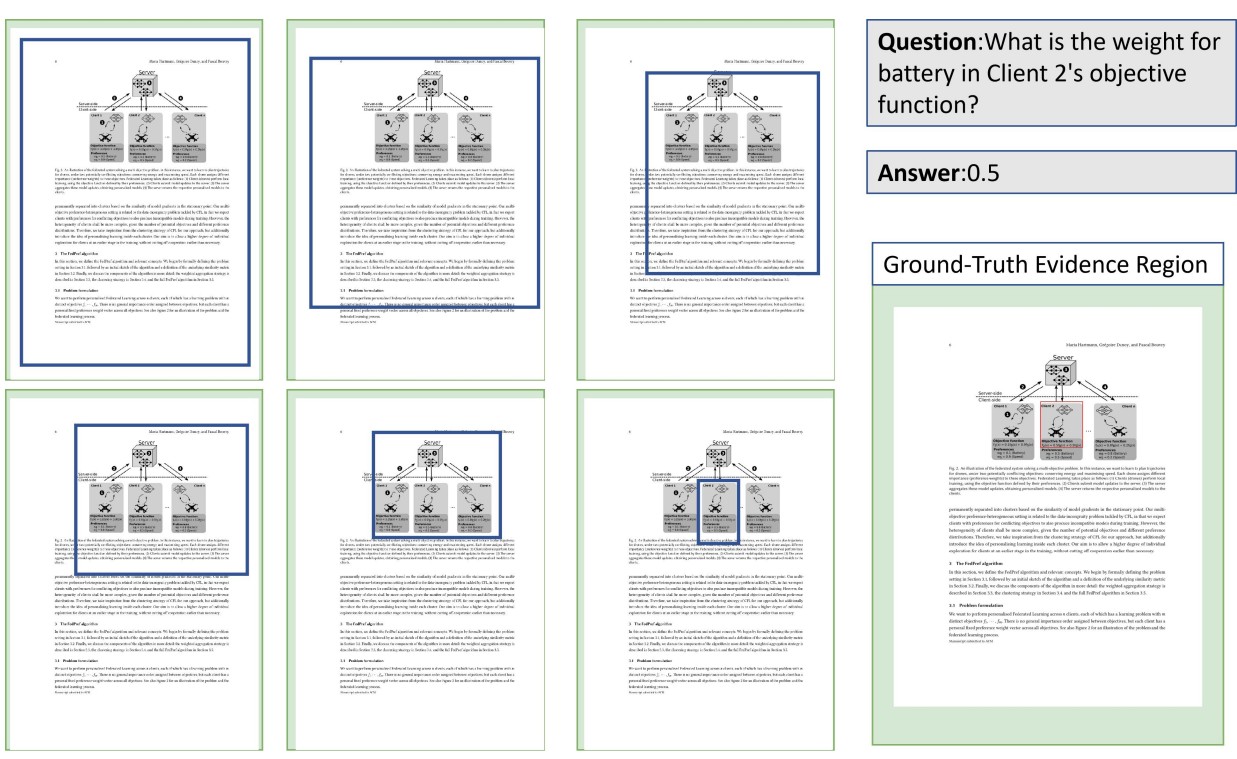

*Figure 8.* Progressive Evidence Region Localization Example

