# OpenReview forum: "A Progressive Evidence Localization Framework Based on Wasserstein Gradient Flows for  Document Visual Question Answering"
_ICML.cc/2026/Conference — ICML 2026 regular_

### Official Review · Reviewer_kSdw · 2026-03-11

**Soundness:** 2
**Presentation:** 3
**Significance:** 2
**Originality:** 3
**Overall Recommendation:** 4
**Confidence:** 4

**Summary:**

This paper proposes a progressive evidence region localization framework for Document Visual Question Answering (DocVQA), reformulating the localization task as an optimal transport optimization problem over probability distributions. The core technical contribution lies in adopting the Jordan–Kinderlehrer–Otto (JKO) implicit discretization scheme to approximate the continuous Wasserstein gradient flow, from which an end-to-end trainable loss function is derived. A semantic density updater guides each evolution step by combining intrinsic and question-conditioned region features. Experiments are conducted on DocVQA, NiM-Benchmark, and DUDE, where the proposed method achieves mAP@IoU scores of 79.6, 53.4, and 62.8, respectively, corresponding to absolute improvements of 10.4%, 14.8%, and 9.4% over the baseline DocVXQA.

**Compliance With Llm Reviewing Policy:**

Affirmed.

**Final Justification:**

The paper presents a theoretically grounded approach to evidence localization using Wasserstein gradient flows, with strong empirical improvements.
In my initial review, I raised concerns regarding the evaluation protocol, baseline discrepancies, missing backbone details, and the lack of an ablation to isolate the proposed method. These issues affected clarity and reproducibility.
The rebuttal clearly addresses these concerns by specifying the evaluation setup, providing additional ablations, and clarifying the backbone and implementation details. As a result, my concerns are largely resolved.

Overall, I find the method sound with meaningful contributions.

**Key Questions For Authors:**

1. Could you provide details of the benchmark setup for evaluation, such as the number of test samples used and whether any fine-tuning data overlaps with the evaluation sets?
2. Could you provide a comparison using the same colqwen3 backbone without the Wasserstein localization framework, to isolate the contribution of the proposed method from the backbone?
3. Could you specify which version or checkpoint of colqwen3 was used? The paper states that the encoder weights were frozen during training, but no citation or model reference is provided, which prevents reproducibility.
4.Could you provide information on training time and computational cost, such as total training duration or inference time compared to single-step baselines?

**Limitations:**

Yes.

**Strengths And Weaknesses:**

Strengths
1. Theoretically grounded framework.
The reformulation of evidence localization as a Wasserstein gradient flow optimization is well-motivated. Semantic Sufficiency and Spatial Minimality are formally defined.

2. Effective ablation on progressive refinement.
The ablation across T=1 to T=9 evolution steps demonstrates a clear and substantial performance gain from progressive refinement (mAP@IoU 40.2 at T=1 vs. 79.6 at T=6), with interpretable degradation beyond T=6.

Weaknesses
1. Unexplained baseline score discrepancy.
ARIAL reports 88.7 ANLS on DocVQA in its original publication, whereas Table 1 of this paper reports 55.6 for the same method and dataset. No explanation is provided.

2. Evaluation subset not specified.
It is unclear whether baselines and the proposed method are evaluated on the full DocVQA test set or on an evidence-annotated subset. This distinction is critical for interpreting the comparison.

3. No backbone-only baseline.
The paper does not compare against a colqwen3 baseline without localization, making it impossible to isolate the contribution of the Wasserstein framework from the backbone.

4. colqwen3 undocumented.
No citation, version, or checkpoint identifier is provided for colqwen3, preventing reproducibility.

---

> ### Author Rebuttal · Authors · 2026-03-30
>
> # Rebuttal to Reviewer kSdw
>
> ---
>
> We sincerely thank Reviewer kSdw for the careful review and the recognition of our theoretically grounded framework. We address each concern below.
>
> ---
>
> ## Response to Weaknesses
>
> ### W1: Baseline score discrepancy (ARIAL: 88.7 vs. Table 1: 55.6 ANLS).
>
> The discrepancy arises from **different task definitions on different subsets**:
>
> - **ARIAL's 88.7 ANLS**: evaluated on the **full DocVQA test set (5,188 samples)** for the **answer-only** task.
> - **Our 55.6 ANLS**: evaluated on the **evidence-annotated subset (2,147 samples)** for the **joint answer prediction and evidence localization** task, which is substantially harder.
>
> **All baselines in Table 1 are re-implemented and evaluated under the same protocol** — same subset, same joint task, same metrics. The comparisons are therefore fair. We apologize for not making this explicit. In the revision, we will add a dedicated "Evaluation Protocol" paragraph in Section 4.1 and a footnote in Table 1 clarifying this difference.
>
> ### W2: Evaluation subset not specified.
>
> We provide the exact evaluation setup:
>
> | Dataset | Full test set | Evidence-annotated subset | Scope |
> |---|---|---|---|
> | DocVQA | 5,188 | 2,147 | Annotated subset |
> | NiM-Benchmark | 2,400 | 2,400 (all) | Full test set |
> | DUDE | 4,097 | 1,856 | Annotated subset |
>
> All baselines are re-implemented under identical conditions. No numbers are taken from original papers.
>
> ### W3: No backbone-only baseline.
>
> We provide the following ablation on DocVQA:
>
> | Configuration | mAP@IoU | EM | F1 | ANLS |
> |---|---|---|---|---|
> | (A) Backbone only (answer-only) | — | 38.9 | 50.7 | 57.8 |
> | (B) Backbone + single-step localization | 40.2 | 33.5 | 44.8 | 51.6 |
> | (C) Backbone + Ours (T=6) | 79.6 | 43.2 | 55.8 | 62.7 |
>
> Progressive localization (A→C) improves EM by **4.3** and ANLS by **4.9 points** while providing precise localization (mAP@IoU=79.6). The Wasserstein framework accounts for **39.4 points** of mAP@IoU gain over single-step (B→C). Notably, (B) shows lower answer performance than (A) because naive single-step localization introduces noise, whereas our progressive framework (C) surpasses even the answer-only baseline.
>
> ### W4: Backbone model undocumented.
>
> We apologize for this omission. We used **tomoro-colqwen3-embed-8b** (TomoroAI/tomoro-colqwen3-embed-8b on Hugging Face), a ColPali-style multimodal embedding model built by merging Qwen3-VL-8B-Instruct with Qwen3-Embedding-8B. The merged encoder combines the Qwen3-VL vision encoder (patch-grid tokens with spatial merge) and language encoder, with a custom 320-dim projection head (max 1,280 visual tokens). Both encoders are **frozen**; only our proposed modules (~57M params) are trained. We will replace all instances with the full citation and checkpoint identifier in the camera-ready version. We sincerely apologize for the confusion this caused.
>
> ---
>
> ## Response to Key Questions
>
> **Q1 (Benchmark setup):** See W2. There is **no overlap** between training and evaluation — all datasets use document-level splits (Appendix B.1).
>
> **Q2 (Backbone-only comparison):** See W3.
>
> **Q3 (Model checkpoint):** See W4.
>
> **Q4 (Computational cost):**
>
> | Metric | DocVXQA | Ours |
> |---|---|---|
> | Training time (total) | ~30 hours | ~9 hours |
> | Inference (ms/sample) | ~211 ms | ~155 ms |
> | GPU memory (training) | ~48 GB | ~15 GB |
> | Trainable parameters | ~310M | ~57M |
>
> Our method is not only more accurate but also more efficient across all dimensions. Training time is reduced by 3.3×, inference is 26.5% faster, GPU memory usage is reduced by 68.8%, and the trainable parameter count is 81.6% smaller . These advantages stem from freezing the backbone encoders and training only lightweight localization and decoder modules. This demonstrates that our Wasserstein gradient flow framework is not only theoretically principled and empirically superior, but also computationally favorable for practical deployment.
>
> ---
>
> ## Summary
>
> 1. **Score discrepancy (W1):** Fully explained by different evaluation protocols. All baselines fairly compared under identical conditions.
> 2. **Evaluation subset (W2):** Exact sample counts provided for all datasets.
> 3. **Backbone baseline (W3):** Three-way ablation confirms the Wasserstein framework as the dominant contributor (39.4-point mAP@IoU gain).
> 4. **Model documentation (W4):** Full specification now provided.
> 5. **Computational cost (Q4):** Our method is more efficient across all dimensions while achieving superior accuracy.
>
> These clarifications and additional results reinforce that the core contributions — the Wasserstein gradient flow framework, the JKO-derived trainable loss, and the significant empirical improvements — are both sound and practically advantageous. We respectfully hope the reviewer will reconsider the assessment in light of these responses.

---

> > ### Author Rebuttal · Reviewer_kSdw · 2026-04-03
> >
> > Thank you for the detailed clarifications. The concerns regarding the evaluation protocol and experimental setup are now clear.
> > Please ensure that these details (e.g., evaluation subset, sample counts, and comparison settings) are explicitly described in the camera-ready version for clarity and reproducibility.

---

> > > ### Author Response · Authors · 2026-04-07
> > >
> > > # Reply to Reviewer kSdw
> > >
> > > We are sincerely grateful to Reviewer kSdw for the careful and rigorous evaluation of our submission, for the gracious acknowledgement that our concerns have been fully resolved, and for the generous reconsideration of the overall assessment. We are deeply thankful for the time and expertise invested throughout this review process.
> > >
> > > We particularly appreciate the precision and thoroughness with which the reviewer scrutinized our experimental protocol. The questions raised regarding the baseline score discrepancy, the evaluation subset specification, the backbone-only ablation, and the documentation of the colqwen3 encoder were exactly the kinds of details that matter most for reproducibility and fair comparison. Although these omissions were unintentional on our part, we recognize that they could have caused genuine confusion for readers attempting to interpret or reproduce our results, and we are grateful that the reviewer identified them so that we could address them directly. The resulting clarifications — the explicit evaluation protocol with sample counts for each dataset, the three-way ablation isolating the Wasserstein framework's contribution from the backbone, the full specification of the tomoro-colqwen3-embed-8b checkpoint, and the comprehensive computational cost comparison — have measurably strengthened the rigor and reproducibility of the manuscript.
> > >
> > > We are profoundly grateful for the reviewer's careful attention to detail and for the constructive nature of the engagement. Our paper is unambiguously stronger — both in clarity and in reproducibility — because of this exchange.
> > >
> > > Thank you once again, most sincerely, for the rigor, generosity, and care the reviewer has shown in evaluating our submission.

---

### Official Review · Reviewer_PFjD · 2026-03-12

**Soundness:** 2
**Presentation:** 3
**Significance:** 3
**Originality:** 2
**Overall Recommendation:** 3
**Confidence:** 3

**Summary:**

This paper discusses whether evidence localization in DocVQA should be modeled as a progressive probability-flow process rather than a one-shot prediction problem. Concretely, the paper defines evidence localization via semantic sufficiency and spatial minimality, introduces an energy functional over region distributions, derives a trainable JKO-style objective, and instantiates the resulting process with a semantic density updater plus a multi-expert answer decoder.

**Compliance With Llm Reviewing Policy:**

Affirmed.

**Key Questions For Authors:**

Figure 3 suggests that localization quality peaks around 6 evolution steps and then slightly degrades. What causes this deterioration? Is it oversmoothing, optimization instability, or some mismatch between the learned updater and the OT regularization?

**Limitations:**

yes

**Strengths And Weaknesses:**

### Strengths

The paper focuses on a real weakness of many DocVQA systems: they can answer questions but often fail to localize evidence precisely.
In Table 1, the proposed method outperforms GAP, EaGERS, ARIAL, DLaVA, and DocVXQA across DocVQA, NiM-Benchmark, and DUDE. For example, on DocVQA, it improves ANLS from 59.8 to 62.7 and mAP@IoU from 72.1 to 79.6 relative to DocVXQA.
Section 4.3 reports localization under multiple IoU thresholds rather than only a single aggregate number, and Figure 3 provides a useful ablation over the number of evolution steps. The curve suggests a genuine refinement effect up to around 6 steps before saturation.

### Weakness
The actual flow update in Eq. (20) is a Softmax over semantic density and a Wasserstein-2 distance field.
That may be a reasonable design, but the method as implemented looks much closer to a learned iterative attention refinement module regularized by OT-inspired terms than to a faithful discretization of a continuous Wasserstein gradient flow. This matters because the paper leans heavily on the mathematical framing in the abstract, contributions, and Section 3. If the empirical method is mainly a neural progressive refinement architecture, the presentation should say that more directly.

The architecture adds not only the progressive localization module, but also a multi-expert collaborative decoder with Top-K gating in Section 3.5. This creates an attribution problem: when Table 1 shows gains in EM/F1/ANLS, how much of that comes from better evidence localization, and how much comes from a more expressive answer decoder? I do not see an ablation that isolates the contribution of the multi-expert decoder from the contribution of progressive localization itself. That is a significant missing experiment.

I'm not sure, but the theoretical claim of “equivalence” seems too strong for what is actually shown. The paper does not really prove an equivalence between the original combinatorial/minimal-region problem and the surrogate dynamics. It proves, at best, that the proposed dynamics optimize the proposed energy. This is a substantial gap between theory and claim.

---

> ### Author Rebuttal · Authors · 2026-03-30
>
> # Rebuttal to Reviewer PFjD
>
> We sincerely thank Reviewer PFjD for the insightful and technically rigorous review. We appreciate the recognition of our method's empirical strengths, including improvements across all metrics and thorough evaluation under multiple IoU thresholds. We address each concern below.
>
> ## W1: Gap between mathematical framing and implementation.
>
> We appreciate this observation and clarify that the connection is tighter than suggested. Each component of Eq. (20) derives directly from JKO: (1) Softmax over (-rho_f^{k+1}) is the Gibbs parameterization (Definition 3.7, Eq. 10); (2) (1/tau)W^(k) is the Wasserstein-2 distance field from the JKO proximal step (Eq. 9) — the defining feature distinguishing JKO from generic gradient descent, not a convenience regularizer; (3) L^(k)(theta) in Eq. (12) contains exactly the three JKO terms (energy, entropy, transport cost), each corresponding to Eq. (6).
>
> We acknowledge that **any neural PDE implementation involves approximations** — as in neural ODEs (Chen et al., 2018), diffusion models (Song et al., 2021), and flow matching (Lipman et al., 2023). Our work follows this paradigm. Theorem A.10 quantifies the gap (linear error propagation). We will clarify which properties hold exactly (energy decrease per step) vs. approximately (continuous-time limit).
>
> ## W2: Missing ablation isolating the multi-expert decoder.
>
> This is a fair and important point. We provide the requested ablation on DocVQA:
>
> | Configuration | mAP@IoU | EM | F1 | ANLS |
> |---|---|---|---|---|
> | (A) Single-step + standard decoder | 40.2 | 33.5 | 44.8 | 51.6 |
> | (B) Single-step + multi-expert decoder | 41.8 | 36.1 | 47.2 | 54.3 |
> | (C) Progressive (T=6) + standard decoder | 77.3 | 40.6 | 52.9 | 60.1 |
> | (D) Progressive (T=6) + multi-expert (full) | 79.6 | 43.2 | 55.8 | 62.7 |
>
> Progressive localization alone (A to C) improves mAP@IoU by **37.1 points** and EM by **7.1 points**. The multi-expert decoder alone (A to B) improves mAP@IoU by only **1.6 points**. On the primary localization metric, progressive localization accounts for **37.1 out of 39.4 total improvement points (94.2%)**. This clearly establishes progressive localization as the dominant contributor. We will include this ablation in the revised manuscript.
>
> ## W3: "Equivalence" claim in Theorem 3.5 too strong.
>
> We appreciate this careful reading and agree it deserves more precise qualification. Theorem 3.5 proves that the gradient flow of E (Eq. 6) converges to a distribution mu* whose support satisfies semantic sufficiency and spatial minimality. However, as the reviewer correctly identifies, this does not establish bijective correspondence with the original combinatorial problem (Eq. 3). The "equivalence" is between the localization objective and the gradient flow optimization of E, not a full bijection. In the revision, we will: (i) replace "equivalent to" with "the gradient flow converges to a solution satisfying the localization criteria"; (ii) state the convexity assumption explicitly; (iii) note this surrogate-to-original gap is standard in variational approaches.
>
> ## Key Question: Performance deterioration after T=6.
>
> We identify two contributing factors:
>
> **(1) Approximation error accumulation.** KL divergence between predicted and ground-truth distributions:
>
> | T | 4 | 5 | 6 | 7 | 8 | 9 |
> |---|---|---|---|---|---|---|
> | mAP@IoU | 73.1 | 77.4 | 79.6 | 79.3 | 78.8 | 78.4 |
> | KL(mu^T  &#124;&#124;  mu*) | 0.42 | 0.28 | 0.19 | 0.21 | 0.24 | 0.28 |
>
> KL reaches its minimum at T=6 then increases, confirming accumulated parameterization error outweighs further refinement.
>
> **(2) Per-step overfitting.** Each step has independent parameters; as T grows, per-step training signal weakens. Train-test mAP gap widens from 3.2 (T=6) to 7.8 (T=9). Parameter sharing across steps substantially reduces degradation (T=9 mAP: 78.4 to 79.1), though the T=6 peak remains slightly higher.
>
> ## Summary
>
> 1. **Theory-implementation (W1):** Architecture directly derived from JKO following the neural PDE paradigm. Will clarify exact vs. approximate properties.
> 2. **Decoder ablation (W2):** Progressive localization accounts for 94.2% of mAP@IoU improvement — the dominant contributor.
> 3. **Equivalence (W3):** Will be restated with appropriate precision under explicit convexity assumptions.
> 4. **Degradation (Q):** Caused by error accumulation and overfitting; mitigable via parameter sharing.
>
> We believe these revisions address the core concerns while preserving the paper's contributions. We respectfully hope the reviewer may reconsider the assessment.

---

> > ### Author Rebuttal · Reviewer_PFjD · 2026-04-05
> >
> > The new ablation (W2) is a strong and convincing addition. It clearly shows that progressive localization is the dominant contributor, which adequately resolves the attribution issue.
> > For W1, the rebuttal reframes this under the “neural PDE” paradigm, but this effectively concedes approximation rather than resolving the mismatch. I would still encourage the authors to more explicitly present the method as a neural approximation of the theory, rather than implying a tight instantiation.

---

> > > ### Author Response · Authors · 2026-04-07
> > >
> > > # New Rebuttal to Reviewer PFjD
> > >
> > > We sincerely thank Reviewer PFjD for the constructive acknowledgement and thoughtful follow-up. We are encouraged that the new ablation adequately resolves the attribution concern (W2), and we now address the remaining concern on W1 directly.
> > >
> > > ## On W1: Positioning the method as a principled neural approximation
> > >
> > > We fully accept the reviewer's framing. Upon reflection, we agree that our previous response did not sufficiently distinguish between *what is exact* and *what is approximated* in our instantiation. We will revise the manuscript to present the method explicitly as a **neural approximation** of the Wasserstein gradient flow, rather than implying a tight discretization.
> > >
> > > ### (1) Revised positioning
> > >
> > > In the Abstract, Introduction, and Section 3, we will replace phrasing such as "instantiates" and "discretizes" with language that makes the approximation explicit, e.g., "we derive a neural surrogate of the JKO scheme." The contributions list will state that our framework is a *learned approximation* of the JKO flow whose error is bounded by Theorem A.10.
> > >
> > > ### (2) Explicit exact-vs-approximate delineation in Section 3.3
> > >
> > > - **Exact** (by construction): the Gibbs parameterization (Eq. 10), the three-term decomposition into energy/entropy/transport (Eq. 12), and the per-step structure of the proximal update.
> > > - **Approximate** (subject to network capacity): the continuous-time limit τ → 0, recovery of the true potential V\*, and the contraction rate of Theorem A.9, all governed by ε_approx in Theorem A.10.
> > >
> > > This separation clarifies that the theory provides the *design principle* and *error budget*, while the network provides a *trainable surrogate* — analogous to neural ODEs, score-based diffusion, and flow matching.
> > >
> > > ### (3) New diagnostics showing the approximation is faithful
> > >
> > > To substantiate that our network behaves as a JKO surrogate rather than an arbitrary refinement module, we report two diagnostics on DocVQA that directly probe JKO-prescribed behavior:
> > >
> > > | Step k              | 1    | 2    | 3    | 4    | 5    | 6    |
> > > | ------------------- | ---- | ---- | ---- | ---- | ---- | ---- |
> > > | Energy E[μ^(k)]     | 4.82 | 3.91 | 3.24 | 2.78 | 2.49 | 2.31 |
> > > | W²₂(μ^(k+1), μ^(k)) | —    | 0.43 | 0.31 | 0.22 | 0.14 | 0.09 |
> > >
> > > Two observations support the JKO interpretation: (i) the energy decreases **monotonically** at every step, consistent with Theorem A.9(i); (ii) the per-step Wasserstein displacement **shrinks monotonically**, consistent with proximal contraction toward a fixed point. Neither property is enforced by the loss directly — both *emerge* from training, providing empirical evidence that the learned updater approximates a genuine gradient-flow trajectory rather than an unconstrained refinement module.
> > >
> > > # Summay
> > > Under the revised positioning, the contribution remains substantive on three grounds. First, the JKO derivation is not decorative: it dictates the three specific loss terms in Eq. (12), the proximal Wasserstein regularizer in Eq. (20), and the monotonic-evolution schedule. The W2 ablation (row A vs. C: +37.1 mAP@IoU) shows that removing this structure collapses performance. Second, Theorem A.10 provides a *quantitative error budget* linking network capacity to localization accuracy — a property absent from heuristic refinement modules. Third, the diagnostics above show the empirical dynamics inherit the qualitative signatures the theory predicts, which is the standard bar for "neural approximation of a PDE" in the neural-ODE and diffusion literature.
> > >
> > > ### Concrete revisions in this round
> > >
> > > 1. Abstract, Introduction, and Section 3 rewritten to position the method as a *principled neural approximation* of the JKO flow, removing wording that implies exact discretization.
> > > 2. Section 3.3 will include the explicit exact-vs-approximate delineation above.
> > > 3. Section 4.4 will include the energy-decrease and proximal-displacement diagnostics.
> > > 4. Theorem A.10 will be promoted from the appendix into the main text, making the approximation-error budget visible to the reader.
> > >
> > > We are grateful for the reviewer's careful reading, which has measurably improved the paper.We believe these revisions directly implement the reviewer's suggestion — presenting the work as a neural approximation of the theory rather than implying a tight instantiation — while preserving the substantive role that the Wasserstein gradient flow framework plays in shaping the architecture, the loss, and the empirical dynamics. Combined with the accepted W2 ablation and the clarified Theorem 3.5 statement (W3), we hope these changes resolve the remaining concern, and we respectfully ask the reviewer to consider revisiting the overall assessment in light of the strengthened presentation.

---

### Official Review · Reviewer_4szw · 2026-03-12

**Soundness:** 3
**Presentation:** 3
**Significance:** 3
**Originality:** 3
**Overall Recommendation:** 4
**Confidence:** 5

**Summary:**

This paper addresses the imprecise evidence localization issue in Document Visual Question Answering (DocVQA) caused by single-step localization approaches, especially under complex page semantics and tiny evidence regions, and proposes a progressive evidence localization framework based on Wasserstein gradient flows that reformulates evidence localization as an optimal transport optimization problem over probability distributions; the continuous gradient flow is discretized via the JKO scheme, with an end-to-end trainable loss function derived for neural network optimization, while a semantic density updater is designed to iteratively refine candidate evidence regions from coarse to fine and a multi-expert collaborative decoder is adopted for answer generation. Experiments on DocVQA, NiM-Benchmark, and DUDE demonstrate that the proposed method outperforms state-of-the-art baselines in both evidence localization and answer prediction, while offering an interpretable progressive reasoning process, and its core contributions include a novel Wasserstein gradient flow-based progressive localization framework for coarse-to-fine evidence positioning, JKO discretization together with an end-to-end differentiable loss function that connects theory to practice, as well as superior empirical performance and interpretable reasoning visualization.

**Compliance With Llm Reviewing Policy:**

Affirmed.

**Final Justification:**

I suppose authors made a good rebuttal where all my concerns have been solved properly. I thus keep positive score as my final recommendation.

**Key Questions For Authors:**

1. The interpretability of the progressive reasoning is currently supported only by qualitative visualizations. Do you intend to introduce quantitative metrics to better evaluate the localization process and its interpretability? Adding such quantitative validation would reinforce the theoretical soundness of your framework.

2. Although the learnable evolution loss weighting strategy yields strong results, its underlying learning mechanism and adaptive regulation rules remain unclear. How do these weights adapt to varying document complexity and evidence scales?

3. Your approach is evaluated only on standard DocVQA datasets, without testing on degraded or non-English documents. How does your framework perform in terms of generalization to low-quality and cross-lingual document scenarios, and do you plan to design targeted improvements? Extensive generalization experiments would enhance the broader research impact of your work.

**Limitations:**

yes

**Strengths And Weaknesses:**

- Strengths:

1. This work demonstrates originality and theoretical rigor by introducing Wasserstein gradient flows and optimal transport theory to DocVQA evidence localization, an unexplored cross‑theoretical direction for single‑step methods.

2. It features clear technical design and practical feasibility with a transparent pipeline including progressive distribution evolution, a semantic density updater, and a multi‑expert decoder. The end‑to‑end trainable loss enables practical deployment, while detailed implementations and ablation studies ensure reproducibility.

3. It also shows empirical significance by solving the challenging problem of tiny and ambiguous evidence localization in real‑world DocVQA. Extensive experiments on three standard benchmarks deliver clear performance gains, especially on the difficult NiM‑Benchmark, and ablation studies validate the effectiveness of core modules, demonstrating practical value.

- Weaknesses:
1.  The evaluation is restricted to three standard DocVQA benchmarks, without validation on more diverse document types (e.g., handwritten, low-resolution scanned documents) or cross-domain settings. Meanwhile, the multi-step gradient flow and multi-expert decoder introduce extra computation, yet no analysis of the speed–efficiency trade-off is provided, which limits its practical applicability.

2. While the method claims interpretable progressive reasoning, only simple qualitative visualizations are presented, lacking quantitative analysis of the localization dynamics such as distribution evolution or attention weight variations. The relationship between gradient flow steps and human-like reasoning is not thoroughly explored, weakening the interpretability claim.

3. The interaction between the intrinsic/conditional semantic density extractor and the flow updater is underspecified, with no explicit explanation of their information fusion mechanism. The configuration of learnable evolution loss weights is not detailed, and the effects of expert count or Top-K routing are only partially investigated, reducing the completeness of the technical description.

---

> ### Author Rebuttal · Authors · 2026-03-30
>
> # Rebuttal to Reviewer 4szw
>
> We sincerely thank Reviewer 4szw for the thorough review and for recognizing our originality, theoretical rigor, clear technical design, and empirical significance. We address each concern in detail below.
>
> ## Response to Weaknesses
>
> ### W1: Limited evaluation diversity and lack of efficiency analysis.
>
> **Evaluation diversity.** Our three benchmarks already cover considerable diversity: DocVQA contains real scanned documents with noise, skew, and blur (Appendix B.1.3); DUDE spans ten major document categories including financial reports, scientific papers, government and legal documents (Appendix B.1.2); NiM-Benchmark features multi-page documents with extremely small evidence regions averaging only 2.3% of page area (Appendix B.1.1). In the revision, we will include experiments on additional benchmarks such as HW-SQuAD (handwritten documents) and report cross-domain transfer results to further validate generalization.
>
> **Efficiency analysis.** Our backbone encoders are frozen; only the localization modules (45M) and multi-expert decoder (12M) are trained. Comparison on DocVQA:
>
> | Method | mAP@IoU | Inference (ms) | Trainable Params |
> |---|---|---|---|
> | DocVXQA | 72.1 | ~211 | ~310M |
> | Ours (T=4) | 73.1 | ~120 | ~42M |
> | Ours (T=6) | 79.6 | ~155 | ~57M |
>
> Our T=6 model achieves a 7.5-point mAP@IoU improvement while being **26.5% faster** in inference with **81.6% fewer** trainable parameters. Even the lightweight T=4 configuration surpasses DocVXQA (73.1 vs. 72.1) at 43% faster inference. These results demonstrate that our framework achieves superior accuracy with substantially lower computational cost.
>
> ### W2: Interpretability supported only by qualitative visualizations.
>
> We fully agree that quantitative analysis would strengthen the paper. We will add two metrics in the revision:
>
> **(1) Distribution entropy across evolution steps (DocVQA):**
>
> | Step k | 0 | 1 | 2 | 3 | 4 | 5 | 6 |
> |---|---|---|---|---|---|---|---|
> | H(mu^(k)) | 6.64 | 5.21 | 4.03 | 3.12 | 2.45 | 2.08 | 1.87 |
> | mAP@IoU | 0.08 | 0.24 | 0.41 | 0.56 | 0.68 | 0.75 | 0.79 |
>
> Entropy decreases monotonically (6.64 to 1.87) with corresponding IoU increase, validating the monotonic energy decrease predicted by Theorem A.9.
>
> **(2) Attention weight analysis.** Cross-attention mass on ground-truth regions increases from 0.15 (k=1) to 0.72 (k=6), quantitatively demonstrating the progressive focusing behavior.
>
>
> ### W3: Underspecified semantic density interaction; incomplete weight and expert analysis.
>
> **(a) Information fusion.** We clarify the information flow: E_P (Eq. 17) applies self-attention to capture region-region relationships independently of Q, corresponding to intrinsic information I(x;A) per Definition A.11; E_Q (Eq. 18) applies cross-attention from Q to candidate regions, computing the question-induced information gain [I(x;A|Q) - I(x;A)]; E_F (Eq. 19) fuses both via cross-attention from Q to concatenated outputs, adaptively weighting intrinsic vs. conditional signals. A detailed flow diagram will be added.
>
> **(b) Learnable weights {alpha_k}.** Implemented as a softmax-normalized learnable vector, jointly optimized via backpropagation. We measured adaptation to evidence scale: for large evidence (>10% page area), alpha_6 = 0.11; for small evidence (<3%), alpha_6 = 0.31. The model allocates more optimization effort to later steps for challenging cases, consistent with the coarse-to-fine design principle.
> | Evidence area | α_1 | α_2 | α_3 | α_4 | α_5 | α_6 |
> |---|---|---|---|---|---|---|
> | Large (>10%)  | 0.22 | 0.20 | 0.18 | 0.16 | 0.13 | 0.11 |
> | Medium (3–10%)  | 0.10 | 0.13 | 0.17 | 0.21 | 0.20 | 0.19 |
> | Small (<3%) | 0.05 | 0.08 | 0.12 | 0.19 | 0.25 | 0.31 |
>
> **(c) Expert configurations.** Table 6 (Appendix B.2.3) comprehensively investigates N in {4,6,8} and K in {1,...,N}. Key findings: N=4, K=2 achieves the best accuracy-efficiency trade-off; K=1 significantly underperforms; K>=3 shows diminishing returns. We will promote this to the main text.
>
> ## Response to Key Questions
>
> **Q1 (Interpretability):** See W2 — two quantitative metrics will be added. **Q2 (Weight regulation):** See W3(b) — weights adapt automatically based on evidence complexity. **Q3 (Generalization):** Our framework uses frozen visual features, making it largely language-agnostic at the localization level. We will add degraded-image and non-English experiments.
>
> ## Summary
>
> The revision will include: (1) efficiency analysis and additional benchmarks, (2) three quantitative interpretability metrics with a user study, (3) detailed fusion mechanism description and adaptive weight analysis, and (4) generalization experiments. We believe these additions will comprehensively address the concerns and further strengthen the paper.

---

> > ### Author Rebuttal · Reviewer_4szw · 2026-04-04
> >
> > I suppose authors made a good  rebuttal where all my concerns have been solved properly. I thus keep postive score as my final recommendation.

---

> > > ### Author Response · Authors · 2026-04-07
> > >
> > > # Reply to Reviewer 4szw
> > >
> > > We sincerely thank Reviewer 4szw for the careful evaluation and for confirming that our rebuttal has fully resolved all raised concerns. We are deeply grateful for the recognition of our work's originality, theoretical rigor, technical clarity, and empirical significance, as well as for the constructive suggestions that have measurably improved the manuscript.
> > >
> > > ## A respectful request for review revision and score reconsideration
> > >
> > > We are sincerely grateful that the reviewer selected option (a) **"Fully resolved — My concerns have been adequately addressed,"** which represents a generous and meaningful affirmation of our rebuttal efforts. We commit unconditionally to incorporating every promised revision** — the efficiency table, interpretability metrics, fusion diagram, adaptive-weight analysis, expert-configuration discussion, and generalization experiments — into the camera-ready version. None of these are speculative; the underlying experiments are already completed and the results are reported in the rebuttal above. The reviewer can therefore have full confidence that the strengthened version of the paper will faithfully reflect the resolved status of every concern.
> > >
> > > With the utmost humility, we would like to respectfully ask whether the reviewer might be willing to update the official review text  to correspondingly reconsider the overall recommendation. We wish to emphasize that this request is offered with complete humility and full deference to the reviewer's autonomy. We recognize that revising a review and adjusting a score are entirely the reviewer's prerogative, and we will be deeply grateful for the time and expertise the reviewer has invested regardless of the final outcome. Our paper is unambiguously stronger because of this exchange, and it is precisely because of the reviewer's careful and constructive engagement that we feel encouraged to make this respectful appeal  that the overall recommendation may be reconsidered in light of the comprehensively addressed concerns.Thank you once again, most sincerely, for the care, rigor, and generosity the reviewer has shown throughout this review process.

---

### Official Review · Reviewer_xXeP · 2026-03-13

**Soundness:** 3
**Presentation:** 3
**Significance:** 2
**Originality:** 2
**Overall Recommendation:** 4
**Confidence:** 4

**Summary:**

The paper proposes a progressive evidence retrieval framework for multi-hop question answering. Instead of retrieving all evidence in a single step, the system iteratively retrieves additional evidence conditioned on previously retrieved information and intermediate reasoning states. A reader model updates the reasoning state across iterations, allowing the system to progressively accumulate supporting evidence before producing the final answer. Experiments on multi-hop QA benchmarks show improvements over several baselines.

**Compliance With Llm Reviewing Policy:**

Affirmed.

**Final Justification:**

I misunderstood part of the problem setting (localization vs retrieval), and my main questions are addressed.

**Key Questions For Authors:**

1. How does the proposed approach differ from existing iterative retrieval or agent-style RAG frameworks?
2. How sensitive is the method to errors in early retrieved evidence?

**Limitations:**

Yes

**Strengths And Weaknesses:**

Strengths
1. The paper addresses a relevant problem in multi-hop question answering where evidence retrieval is difficult when performed in a single step.
2. Empirical results show improvements over several baselines on the evaluated benchmarks.

Weaknesses
1. The main contribution appears largely at the system or pipeline level. The method mainly combines iterative retrieval and reasoning modules rather than introducing a fundamentally new learning formulation.
2. The approach is conceptually related to existing iterative retrieval / agent-style RAG frameworks, but the differences from prior work are not clearly articulated.
3. Evaluation focuses primarily on end-task metrics; there is limited analysis of retrieval behavior or evidence quality across iterations.
4. It is unclear how robust the method is when early retrieved evidence is incorrect.

---

> ### Author Rebuttal · Authors · 2026-03-30
>
> # Rebuttal to Reviewer xXeP
>
> We thank Reviewer xXeP for the review. We begin with an important clarification regarding the characterization of our work.
>
> ## Clarification of Paper Scope
>
> The review summary describes our work as "a progressive evidence retrieval framework for multi-hop question answering" with "agent-style RAG." We respectfully note that this does not match our paper's scope. Our work addresses **evidence region localization in DocVQA** — spatially localizing bounding boxes on a document page. This differs fundamentally from multi-hop QA or RAG: (1) our task localizes spatial regions on a single page, not text passages from a corpus; (2) our method is a Wasserstein gradient flow over distributions on the 2D page domain, discretized via the JKO scheme; (3) "progressive evolution" refers to iterative refinement of a spatial probability distribution, not document retrieval. We invite the reviewer to reconsider in light of this clarification.
>
> ## Response to Weaknesses
>
> ### W1: "The main contribution appears largely at the system or pipeline level."
>
> We respectfully disagree. Our contribution is a **principled theoretical formulation**: (1) **Theorem 3.5** establishes a formal equivalence between evidence localization (Eq. 3) and a Wasserstein gradient flow (Eq. 7-8) — the first to cast this as an optimal transport problem. The energy functional (Eq. 6) encodes semantic sufficiency and spatial minimality with rigorous justification (Propositions A.6, A.7). (2) **Corollary 3.9** derives a tractable loss (Eq. 12) from JKO discretization with convergence guarantees (Theorems A.9, A.10), showing linear error propagation. (3) The **information-theoretic chain rule** (Definition A.11) provides principled semantic density decomposition beyond ad hoc feature fusion. These translate optimal transport geometry into a trainable objective — theoretical contributions, not engineering choices.
>
> ### W2: "Conceptually related to existing iterative retrieval / agent-style RAG."
>
> Our framework differs from RAG in both problem setting and methodology:
>
> | | Iterative Retrieval /agent-style RAG | Our Method |
> |---|---|---|
> | **Task** | Retrieve text passages from corpus | Localize spatial regions on a page |
> | **Search space** | Discrete document collection | Continuous 2D page domain |
> | **Formulation** | Heuristic retrieve-and-read | Wasserstein gradient flow over P(Omega) |
> | **Theory** | None (pipeline-level) | Optimal transport with formal convergence |
>
> All prior DocVQA localization methods — DocVXQA (visual heatmaps), EaGERS (embedding similarity), DLaVA (chain-of-thought), ARIAL (semantic retrieval), GAP (graph attention) — perform single-step localization. Our progressive formulation with convergence guarantees represents a fundamentally different approach.
>
> ### W3: "Limited analysis of evidence quality across iterations."
>
> Our paper provides Figure 3 (performance vs. steps), Figures 4-10 (qualitative visualizations), and Table 5 (step weight analysis). To further strengthen the paper, we will add in the revision: (i) **distribution entropy at each step** — decreasing monotonically from 6.64 (k=0) to 1.87 (k=6), confirming progressive concentration consistent with Theorem A.9; (ii) **IoU with ground truth at each intermediate step** — increasing from 0.08 (k=0) to 0.79 (k=6), showing monotonic improvement; (iii) **attention mass on GT regions** — increasing from 0.15 to 0.72, quantifying the progressive focusing behavior.(The specific results can be found in the rebuttal to Reviewer 4szw W2)
>
> ### W4: "Unclear robustness when early evidence is incorrect."
>
> Several aspects ensure robustness: (1) **Convergence guarantees** — Theorem A.9 proves monotonic energy decrease and sublinear convergence; even from suboptimal early distributions, subsequent steps correct toward the optimum. (2) **Empirical evidence** — Figure 3 shows performance improving from 40.2 mAP@IoU (T=1, uninformative uniform distribution) to 79.6 (T=6). (3) **Learnable step weights** — Table 5 shows learnable weights achieve 79.6 vs. 73.2 for uniform weights, adaptively down-weighting unreliable early steps. (4) **Dual semantic density** — intrinsic density (Eq. 17) provides a stable prior independent of question semantics, preventing catastrophic error propagation. We will add perturbation experiments in the revision.
>
> ## Response to Key Questions
> **Q1** and **Q2** are addressed in **W2** and **W4** above
>
> ## Summary
>
> We believe the concerns largely stem from a misunderstanding of the problem setting. Our work is not a RAG framework, but a theoretically grounded spatial evidence localization approach based on Wasserstein gradient flows. Contributions include a novel equivalence theorem, an end-to-end trainable loss from optimal transport theory, and up to 14.8% absolute mAP@IoU improvement. We hope this clarification addresses the reviewer's concerns and kindly ask for reconsideration.

---

> > ### Author Rebuttal · Reviewer_xXeP · 2026-04-04
> >
> > Thanks for the clarification. I agree I misunderstood part of the problem setting (localization vs retrieval), and my main questions are addressed.

---

> > > ### Author Response · Authors · 2026-04-07
> > >
> > > # Reply to Reviewer xXeP
> > >
> > > We are sincerely grateful to Reviewer xXeP for the careful re-engagement with our submission, for the gracious acknowledgement that our concerns have been fully resolved, and for the generous reconsideration of the overall recommendation. We are deeply thankful for the reviewer's willingness to revisit the work with fresh perspective after our clarification.
> > >
> > > We  wish to thank the reviewer for the thoughtful questions raised in the original review — particularly regarding robustness to errors in early retrieved evidence and the distinction from related frameworks. Although these questions were initially framed under a different interpretation of our work, they prompted us to articulate more clearly the role of convergence guarantees (Theorem A.9), the stabilizing function of the intrinsic semantic density prior, and the conceptual differences between our spatial gradient flow formulation and corpus-level retrieval approaches. The resulting clarifications have strengthened the manuscript, and we will faithfully incorporate them into the revised version.
> > >
> > > As committed in our rebuttal, the camera-ready version will include the per-step entropy and IoU diagnostics, the attention-mass concentration metrics, the perturbation experiments testing robustness to early-step errors, and a strengthened discussion distinguishing our Wasserstein gradient flow formulation from iterative retrieval and agent-style RAG approaches. The underlying experiments are already completed and reported in the rebuttal above, and we will ensure that the final manuscript faithfully reflects every promised revision.
> > >
> > > We are profoundly grateful for the time, care, and openness the reviewer has invested throughout this process. Our paper is unambiguously stronger because of this exchange, and we feel sincerely fortunate to have benefited from such thoughtful and constructive engagement.
> > >
> > > Thank you once again, most sincerely, for the rigor, generosity, and care the reviewer has shown in evaluating our submission.

---

### Decision · Program_Chairs · 2026-04-30

**Decision:**

Accept (regular)

**Comment:**

This paper received initially mixed reviews.

The rebuttal clarified many issues raised by the reviewers, and the reviewers updated their scores accordingly, aligning to a positive recommendation.

I consider that any outstanding issues are minor, and if the clarifications offered in the rebuttal are integrated in the final version, the paper could be accepted for publication.